# A bacterial endophyte exploits chemotropism of a fungal pathogen for plant colonization

Davide Palmieri[1], Stefania Vitale[2,3], Giuseppe Lima [1], Antonio Di Pietro [2,5✉] & David Turrà [2,3,4,5✉]

Soil-inhabiting fungal pathogens use chemical signals released by roots to direct hyphal growth towards the host plant. Whether other soil microorganisms exploit this capacity for their own benefit is currently unknown. Here we show that the endophytic rhizobacterium *Rahnella aquatilis* locates hyphae of the root-infecting fungal pathogen *Fusarium oxysporum* through pH-mediated chemotaxis and uses them as highways to efficiently access and colonize plant roots. Secretion of gluconic acid (GlcA) by *R. aquatilis* in the rhizosphere leads to acidification and counteracts *F. oxysporum*-induced alkalinisation, a known virulence mechanism, thereby preventing fungal infection. Genetic abrogation or biochemical inhibition of GlcA-mediated acidification abolished biocontrol activity of *R. aquatilis* and restored fungal infection. These findings reveal a new way by which bacterial endophytes hijack hyphae of a fungal pathogen in the soil to gain preferential access to plant roots, thereby protecting the host from infection.

[1] Dipartimento Agricoltura, Ambiente e Alimenti, Università degli Studi del Molise, Campobasso, Italy. [2] Departamento de Genetica, Campus de Excelencia Internacional Agroalimentario ceiA3, Universidad de Córdoba, Córdoba, Spain. [3] Dipartimento di Agraria, Università di Napoli Federico II, Portici, Italy. [4] Center for Studies on Bioinspired Agro-enviromental Technology, Università di Napoli Federico II, Portici, Italy. [5]These authors jointly supervised this work: Antonio Di Pietro, David Turrà. ✉email: ge2dipia@uco.es; davturra@unina.it

Root-associated microbes strongly impact plant growth and productivity. Although some are beneficial, others severely affect plant health[1]. The soil-inhabiting fungus *Fusarium oxysporum* infects more than a hundred different crops, causing vascular wilt and death of the plant[2]. Exemplifying its destructive potential, a recent outbreak of a banana pathogenic isolate of *F. oxysporum* f. sp. *cubense* known as tropical race 4 (TR4) is currently threatening to eradicate the world's most important staple fruit crop[3].

The infection process initiates when *F. oxysporum* hyphae in the soil sense class III peroxidases released by the roots to redirect growth toward the host plant. This chemotropic response requires conserved signaling components, including Ste2, a receptor for peptide pheromone alpha, the NADPH oxidase B (NoxB) complex, as well as elements of the cell wall integrity mitogen activated protein kinase (CWI MAPK) pathway[4–6]. Once the fungus reaches the plant root, it penetrates and grows intercellularly in the cortex. Invasive growth on the plant requires the functionally distinct Invasive Growth (IG) MAPK cascade[7–10]. Inside the plant, *F. oxysporum* secretes an array of effector proteins, which promote host colonization[11]. Among these, a functional homolog of the plant regulatory peptide RALF (rapid alkalinization factor) efficiently induces host alkalinization, which in turn stimulates MAPK-driven invasive growth[12,13]. Thus, chemotropism and invasive hyphal growth are essential steps during the early infection stages of *F. oxysporum*.

It has long been known that pathogen infection can be suppressed through the activity of root-associated bacteria[14]. The modes of action of disease suppression include competition for space, nutrients, or microelements, as well as the production of antifungal compounds or priming of plant immunity[15,16]. For example, biocontrol strains of the gram-negative bacterium *Pseudomonas fluorescens* inhibit the growth of *F. oxysporum* and other root-infecting fungal pathogens by producing antifungal secondary metabolites such as 2,4-diacetylphloroglucinol or phenazine, and by secreting iron-chelating siderophores[17–19]. Endophytic bacteria, which reside within plants without causing disease symptoms, also can protect the host against fungal pathogens[20].

The gram-negative rhizobacterium *Rahnella aquatilis* was reported to promote plant growth and to act as a biocontrol agent on different crops[21,22]. In a previous study, co-inoculation of chickpea plants with the fungal pathogen *F. oxysporum* and *R. aquatilis* isolate *Ra36* resulted in efficient protection from vascular wilt disease[23], although the mode of action is currently unknown. In this study, we identify the mechanisms underlying the biocontrol activity of *Ra36* against *F. oxysporum*. By monitoring bacterial movement toward and along *F. oxysporum* hyphae, we find that *Ra36* exploits fungal chemotropism to efficiently reach the roots of the host plant. We further show that *Ra36* induces gluconic acid (GlcA)-mediated extracellular acidification, thereby interfering with alkaline-triggered fungal infection.

## Results and discussion

**Ra36 protects tomato plants from F. oxysporum infection.** To study in detail the interaction between *Ra36* and *F. oxysporum*, we used the well-characterized reference strain *F. oxysporum* f. sp. *lycopersici* 4287 (*Fol*), a tomato pathogenic isolate whose genome has been sequenced[24]. We also generated *Ra36* transformants constitutively expressing the red fluorescent protein RFP to facilitate microscopic observation of bacterial cells. We first asked whether *Ra36* protects tomato from *Fol* infection. Plants whose roots had been pre-inoculated with *Ra36* prior to inoculation with *Fol* showed significantly higher survival rates and contained

less fungal biomass compared with those inoculated only with *Fol*, when grown either on inert substrate or in sterilized or unsterilized natural soil (Fig. 1a, b, f; Supplementary Fig. 1a, b; Supplementary Data 1). The protective effect of *Ra36* against *Fol* infection was concentration-dependent and increased with higher concentrations of bacterial inoculum (Fig. 1a). Fluorescence microscopy combined with qPCR and plating experiments revealed the presence of *Ra36* in the root cortex and xylem vessels, suggesting that *Ra36* successfully colonizes tomato roots (Fig. 1c; Supplementary Fig. 1c, d; Supplementary Movies 1, 2). When inoculated alone, *Ra36* did not cause any mortality in tomato plants (Fig. 1a). We conclude that *Ra36* is a bacterial endophyte of tomato plants that efficiently suppresses vascular wilt disease caused by *Fol*.

**Ra36 inhibits fungal virulence via GlcA secretion.** To understand how *Ra36* prevents *Fol* infection, we first tested whether the bacterium directly inhibits hyphal growth. Co-cultivation experiments conducted on different media failed to reveal an effect of *Ra36* on hyphal growth or colony development of *Fol* (Supplementary Fig. 1e). We next asked whether *Ra36* targets known virulence mechanisms of *Fol* such as chemotropism toward plant roots[6], adhesion to the root surface[7] or invasive hyphal growth, defined by the capacity to penetrate cellophane membranes and to invade living plant tissue[9]. We found that directed hyphal growth of *Fol* toward tomato root exudates was unaffected by pre-inoculation of the roots with *Ra36* (Supplementary Fig. 1f). However, hyphal network formation and root adhesion were markedly decreased (Supplementary Fig. 1g, h). Moreover, penetration of *Fol* through cellophane membranes and invasion of tomato fruit tissue were impaired upon co-inoculation with *Ra36* (Fig. 1d, e). We conclude that *Ra36* inhibits key virulence mechanisms of *Fol* such as hyphal network formation, root adhesion and invasive growth.

Previous work established that these virulence mechanisms are activated at high pH and inhibited at low pH[12]. Moreover, it was shown that *Fol* induces alkalinization of the host plant tissue to increase its infectious potential[12,13]. Here, we found that the inhibitory effect of *Ra36* on invasive growth and pathogenicity of *Fol* could be reversed by increasing the extracellular pH to 7.0, or reproduced in the absence of the bacterium by decreasing the external pH to 5.0 (Fig. 1d; Supplementary Fig. 2a; Supplementary Data 1). These results suggested that the inhibitory activity of *Ra36* on fungal virulence functions is associated with a decrease in external pH. Indeed, *Ra36* caused a marked and progressive acidification in culture media as well as in sterilized or unsterilized soil, either in the absence or presence of *Fol* and/or tomato roots (Fig. 2a–e; Supplementary Fig. 2b–e). As reported previously[12], *Fol* triggered significant alkalinization when grown in the presence of tomato roots, however, this effect was dramatically reversed to acidification upon co-inoculation with *Ra36* (Fig. 2c, d; Supplementary Fig. 2e).

How does *Ra36* acidify the extracellular medium? It was previously reported that root-colonizing bacteria secrete organic acids, principally GlcA, to lower the pH of the rhizosphere, thereby facilitating solubilization of inorganic phosphates and microelements and promoting plant growth[25,26]. GlcA-mediated acidification by beneficial Pseudomonads was recently shown to modulate root immune responses[27]. Here, we detected accumulation of GlcA in culture medium or in soil inoculated with *Ra36*, which was concomitant with extracellular acidification (Fig. 2d, e; Supplementary Fig. 2b, c). The concentration of GlcA measured in culture media reached 40 mM, which exceeds that previously reported in other rhizobacteria[25,28,29]. We next asked whether GlcA-mediated extracellular acidification by *Ra36* is required for

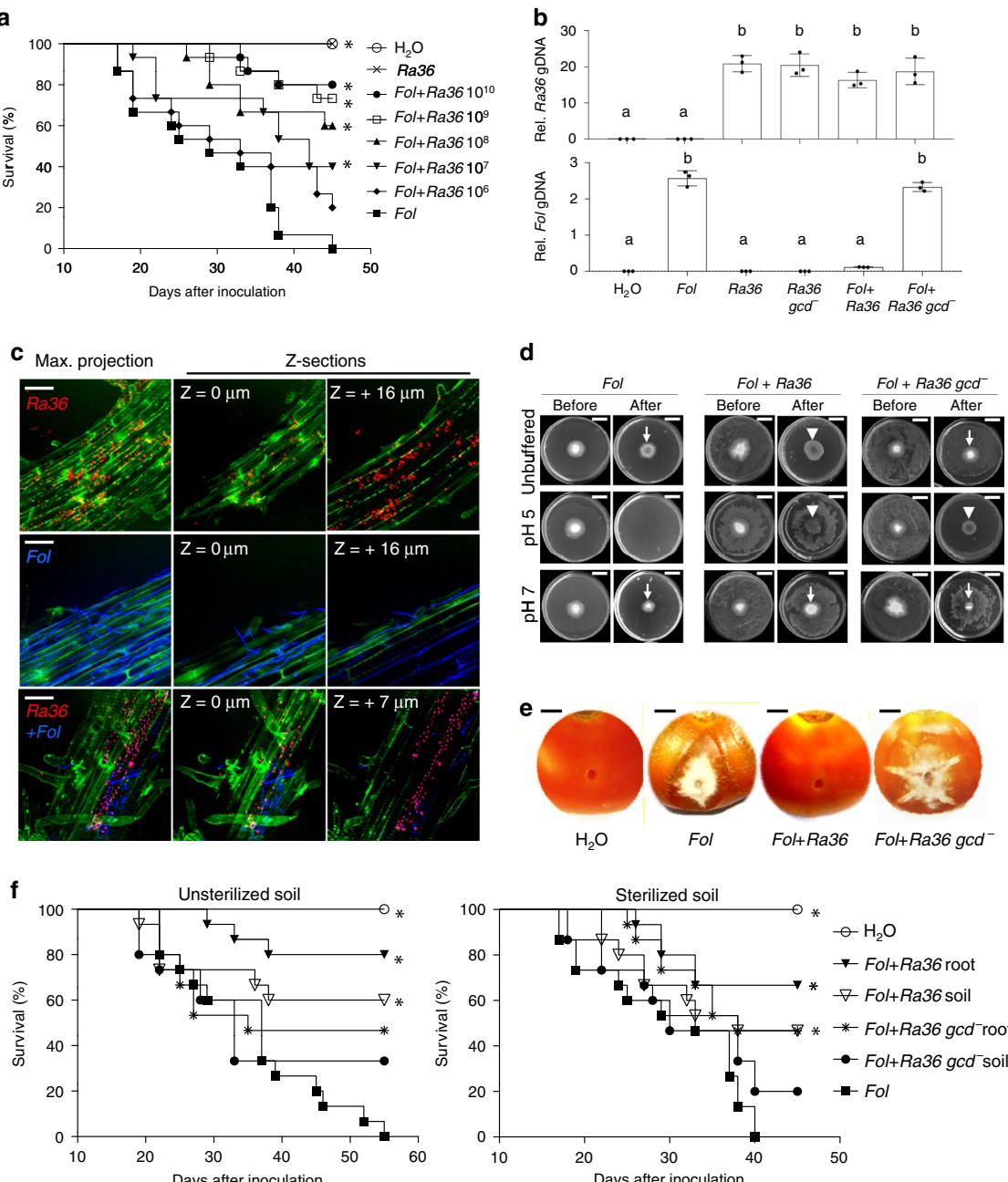

**Fig. 1 *R. aquatilis* inhibits *F. oxysporum* virulence. a** Kaplan–Meier plot showing the survival of tomato plants grown in vermiculite and dip-inoculated or not ($H_2O$) with *Fol*, *Ra36* or *Fol* + *Ra36* at the indicated concentrations (CFU/ml). Number of independent experiments ($n_{i.ex.}$) = 3; 15 plants/treatment. Data shown are from one representative experiment. *$P < 0.05$ versus *Fol* alone according to log-rank test. **b** Relative amount of bacterial and fungal DNA was measured by qPCR of total genomic DNA extracted from the roots of tomato plants grown in vermiculite, 15 d after dip-inoculation with the indicated microorganism(s). $n_{i.ex.}$ = 3; 5 plants/treatment. Data are expressed as ng of target DNA per 100 ng of plant DNA. Data are presented as mean values ±SEM from three independent experiments. Different letters indicate statistically significant differences according to one-way ANOVA followed by Tukey's multiple comparison test ($P < 0.05$). **c** Fluorescence microscopy showing upper, inner, and maximum projections (50 Z-sections) of representative tomato roots stained with calcofluor white (false color green), 48 h after inoculation with RFP-tagged *Ra36* (red) and/or GFP-tagged *Fol* (false color blue). $n_{i.ex.}$ = 3; 2 technical replicates each. Scale bar, 50 μm. **d** Invasive growth of *Fol* on cellophane membranes placed on MMU medium, either unbuffered or buffered at the indicated pH, and lawned or not with *Ra36*. Plates were imaged before and after cellophane removal to visualize *Fol* penetration through the cellophane (arrows). Arrowheads indicate *Ra36* colonies at the center of the plate. $n_{i.ex.}$ = 3, 3 plates each. Scale bar, 2 cm. **e** Invasive growth of *Fol* on tomato fruits, after inoculation with the indicated microorganism(s). $n_{i.ex.}$ = 3, 3 fruits each. Bar, 0.5 cm. **f** Kaplan–Meier plots showing the survival of tomato plants grown in unsterilized or sterilized agricultural soil and inoculated or not ($H_2O$) with the indicated microorganism(s), either mixed with the soil (soil) or dip-inoculated onto the roots (root). $n_{i.ex.}$ = 3, 15 plants/treatment. Data shown are from one representative experiment. *$P < 0.05$ versus *Fol* alone according to log-rank test. Source data from **a**, **b**, **f** are provided as Source Data file.

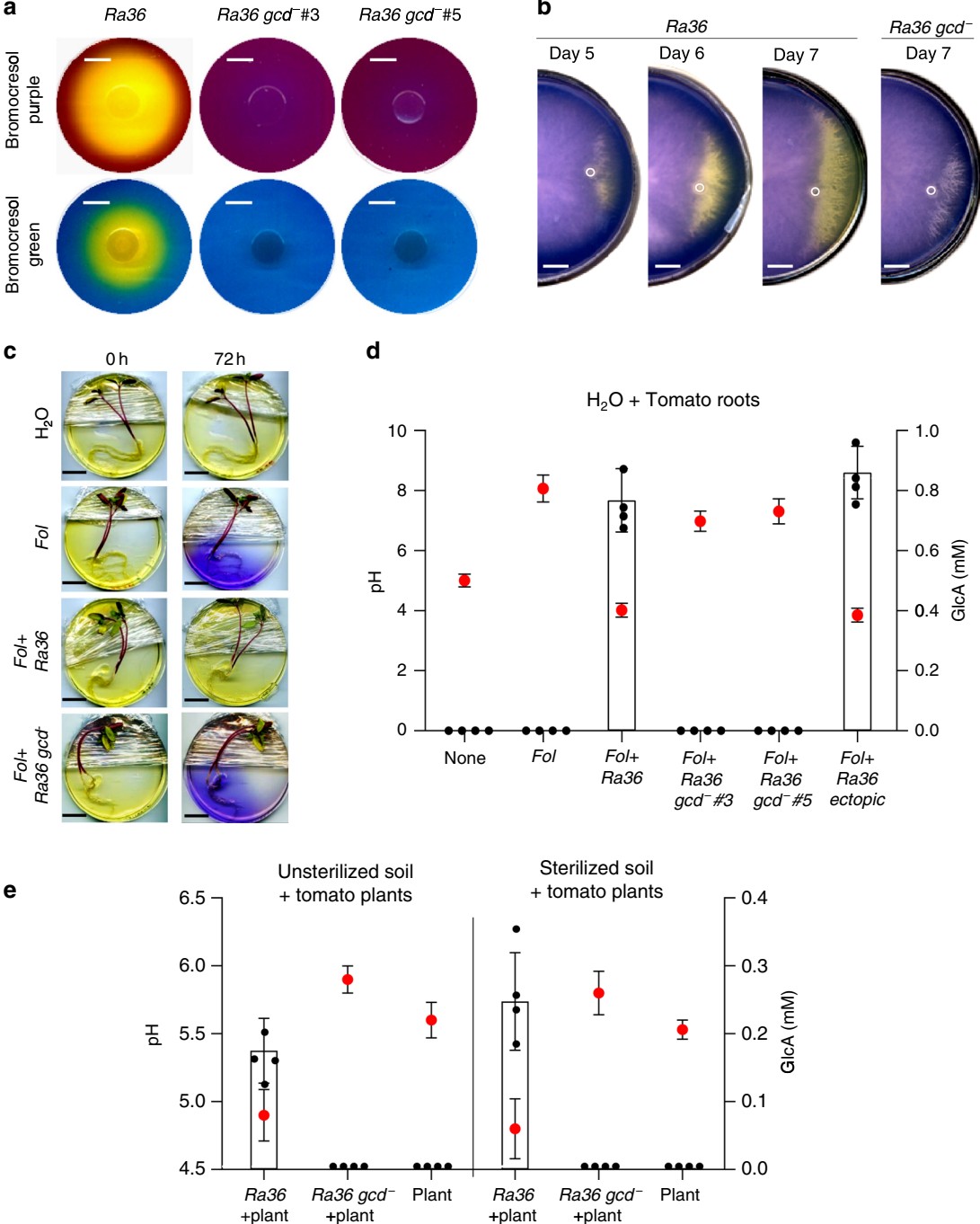

**Fig. 2 R. aquatilis causes gluconic acid-mediated extracellular acidification. a** Representative images showing the indicated *Ra36* strains grown for 2 d on MMU plates (pH 7.0) containing the pH indicators bromocresol green (yellow, pH < 3.8; blue, pH > 5.4) or purple (yellow, pH < 5.2; purple, pH > 6.8). Extracellular acidification is visible as a clear halo. Number of independent experiments ($n_{i.ex.}$) = 4, with two plates each. Scale bar, 1 cm. **b** Representative images illustrating progressive acidification of a *Fol* colony at the indicated times after spot-inoculation (white circles) with *Ra36* but not *Ra36 gcd⁻*, extending toward the margin of the fungal colony. MMU plates (pH 7.0) contain the pH indicator bromocresol purple. $n_{i.ex.}$ = 3, with three plates each. Scale bar, 2 cm. **c** *Ra36* reverses *Fol*-induced host alkalinization. Roots of tomato seedlings were dip-inoculated or not with the indicated *Ra36* strain and placed on water agar containing the pH indicator bromocresol purple (yellow, pH < 5.2; purple, pH > 6.8), either in the absence ($H_2O$) or presence of *Fol* microconidia. Representative plates are shown ($n_{i.ex.}$ = 2, with three plates each). Scale bar, 2 cm. **d** pH (red dots) and gluconic acid (GlcA; bars) measured in water containing a submerged tomato root, either uninoculated ($H_2O$) or 72 h after inoculation with the indicated microorganism(s). $n_{i.ex.}$ = 4, with three replicates each. Data are presented as mean values ± SD from three independent experiments. **e** pH (red dots) and gluconic acid (GlcA; bars) was measured in root exudates of tomato plants grown in unsterilized or sterilized horticultural soil, 2 weeks after inoculation or not with the indicated *Ra36* strain. $n_{i.ex.}$ = 4, with three replicates each. Data are presented as mean values ± SD from three independent experiments. In **a**, **d** *Ra36 gcd⁻* #3 and #5 correspond to independent isogenic *gcd⁻* deletion mutants. Source data from **d**, **e** are provided as a Source Data file.

inhibition of *Fol* virulence functions and plant infection. To test this, we generated isogenic *Ra36 gcd⁻* deletion mutants lacking the predicted orthologue of the enzyme glucose dehydrogenase, which oxidizes D-glucose into GlcA[25,30]. In contrast to the wild-type strain, *Ra36 gcd⁻* mutants failed to secrete GlcA and to acidify the culture medium or the soil, either in the absence or presence of *Fol* or tomato plants (Fig. 2a–e; Supplementary Fig. 2b–e). Importantly, the *Ra36 gcd⁻* mutants failed to inhibit *Fol* virulence functions such as hyphal aggregation, root adhesion, cellophane penetration and tomato fruit invasion (Fig. 1d, e; Supplementary Fig. 1g, h). Moreover, these mutants were significantly less effective than the wild-type *Ra36* strain in protecting tomato plants from *Fol* colonization and wilt disease, either on inert substrate or in sterile or non-sterile soil (Fig. 1b, f; Supplementary Figs. 1a, 4a; Supplementary Data 1). Importantly, GlcA was not required for endophytic growth of *Ra36*, since the *gcd⁻* mutant colonized tomato roots as efficiently as the wild-type strain (Fig. 1b; Supplementary Data 1). Moreover, GlcA had no direct inhibitory effect on *Fol* growth (Supplementary Fig. 2f). Taken together, these results demonstrate that *Ra36* causes extracellular acidification in the presence of tomato roots by secreting GlcA, thereby reverting *Fol*-induced root alkalinization and inhibiting fungal virulence functions and plant infection.

**Ra36 exhibits chemotaxis toward F. oxysporum hyphae.** During the cellophane penetration assay, we noted a progressive accumulation of *Ra36* cells underneath the fungal colony (see arrowheads in Fig. 1d). Because *R. aquatilis* carries flagella for motility[31], we hypothesized that *Ra36* cells exhibit chemotaxis toward *Fol*. To test this, a 2 μl drop of fluorescently labeled *Ra36*-RFP cells was spotted onto soft agarose-coated slides at a distance of 5 mm from a 2 μl drop of fluorescently labeled *Fol*-green fluorescent protein (GFP) germlings. After 30 min, most *Ra36* cells had accumulated at the margin of the bacterial droplet that was oriented toward *Fol*, and subsequently the entire front of the colony moved toward the *Fol* germlings (Fig. 3a). Next, we used a standard capillary assay[32] to quantify chemotaxis and found that *Ra36* exhibited significant chemotactic movement toward sugars (galactose, glucose) and amino acids (tryptophan, glutamine) (Supplementary Fig. 3a; Supplementary Data 1), as well as toward exudates collected either from tomato roots or from *Fol* mycelium (Fig. 3b, c; Supplementary Data 1). Importantly, *Ra36 fliC⁻* deletion mutants lacking the structural flagellin gene *fliC*, which is essential for flagellum function and bacterial motility[33] (Supplementary Fig. 2g), failed to show a chemotactic response toward these stimuli (Fig. 3b; Supplementary Data 1). We further noted that the exudate obtained from tomato roots inoculated with *Fol* had a higher pH (8.0) and elicited a significantly stronger chemotactic response (Fig. 3b; Supplementary Data 1) than that obtained from uninoculated roots (pH 5.8). Interestingly, chemoattraction of *Ra36* cells was abolished when the pH of the *Fol* exudate was artificially lowered from 8.0 to 5.8 to match that of the assay medium, or when the pH of the assay medium was raised from 5.8 to 8.0 to equal that of the *Fol* exudate (Fig. 3c, d; Supplementary Data 1). Moreover, chemoattraction turned into chemorepulsion when the pH of the assay medium was increased to 8.0 and that of the *Fol* exudate was lowered to 5.8 (Fig. 3d; Supplementary Data 1). In line with this, *Ra36* cells maintained at pH 5.8 exhibited positive chemotaxis toward a medium adjusted to pH 8.0, but negative chemotaxis against a medium adjusted to pH 4.0 (Fig. 3e; Supplementary Data 1). The positive or negative chemotactic response was still functional in the *gcd⁻* mutant, but fully abolished in the *fliC⁻* mutant (Fig. 3e). Taken together, these results suggest that *Ra36* exhibits flagellum-driven chemotaxis

toward alkaline pH gradients, such as those produced by *Fol* hyphae growing in the soil or in the tomato rhizosphere.

**Ra36 exploits hyphal chemotropism for root colonization.** Bacterial chemotaxis depends on the free movement of bacterial cells across a continuous layer of liquid, a condition rarely encountered in complex environments such as the soil[34]. Fungal hyphae are known to function as highways promoting the movement of flagellate bacteria across the soil and facilitating their access to nutrient sources[35], thereby increasing bacterial fitness[34,36]. Here, we observed *Ra36* cells that moved along the liquid layer covering *Fol* hyphae and accumulated in the hyphal tip region (Fig. 4a; Supplementary Movies 3, 4). To test the impact of *Fol* hyphae on the ability of *Ra36* to move across air-filled spaces, bacterial and fungal cells were spot-inoculated either individually or in combination on one side of a 5-mm wide groove in the medium, and a tomato root was placed on the opposite side. As expected, *Fol* crossed the gap by means of hyphal bundles and successfully colonized the root, whereas *Ra36* alone failed to do so (Fig. 4b, Supplementary Fig. 3b). However, *Ra36* cells co-inoculated with *Fol* successfully crossed the gap and reached the plant root by navigating along the film of liquid surrounding the fungal hyphae. We further noted a massive accumulation of *Ra36* cells at the tips of leading hyphae as the *Fol* colony approached the root (Fig. 4c, d; Supplementary Movie 5). Upon contact with the root, *Ra36* moved rapidly from the hyphal tip to the root, resulting in progressive evacuation of bacterial cells from the hyphae concomitant with their accumulation on the root surface (Fig. 4c, d; Supplementary Movie 6). Using a capillary chemotaxis assay, we found that *Ra36* cells exposed simultaneously to opposing gradients of exudates from tomato roots and *Fol* hyphae, exhibited significantly higher chemotaxis toward the tomato root exudate (Supplementary Fig. 3c–e; Supplementary Data 1). Together, these results suggest that *Ra36* uses *Fol* to efficiently move through the soil and colonize the roots of the host plant, trading on the ability of the pathogen hyphae to grow across air-filled spaces[37].

*Fol* was previously shown to exhibit directed hyphal growth toward chemoattractants released by tomato roots[6]. Here we found that *Ra36* did not affect chemotropism of *Fol* toward tomato roots or root exudate (Supplementary Figs. 1f, 3f; Supplementary Data 1). We thus speculated that the movement of *Ra36* cells toward the tip of chemotropically growing *Fol* hyphae could result in increased efficiency of bacterial root colonization. To test this idea, wild-type *Ra36* or flagellum-deficient *fliC⁻* cells were point-inoculated at a distance of 5 cm from a tomato plant into natural soil either lacking or containing *Fol*. Strikingly, the roots grown in soil inoculated with *Fol* contained more than tenfold higher biomass of the *Ra36* wild-type strain, but not of the *fliC⁻* mutant, when compared to roots grown in soil lacking *Fol* (Fig. 4e; Supplementary Data 1). In line with this, in contrast to wild-type *Ra36*, the *fliC⁻* mutant failed to protect tomato plants against *Fol* when inoculated into soil, but was as efficient as the wild-type when inoculated directly on the roots (Supplementary Fig. 3g, Supplementary Fig 4a, b). Importantly, no significant increase in bacterial biomass was detected when roots were grown in soil containing the isogenic *Fol ste2Δ* mutant (Fig. 4e; Supplementary Data 1), which is impaired in chemotropic growth toward tomato roots[6]. Thus, *Ra36* significantly benefits from the capacity of *Fol* hyphae to sense and grow toward tomato roots, thereby increasing its efficiency in colonizing the host plant.

Taken together, our results reveal a previously unknown mechanism of a soil-inhabiting bacterial endophyte to gain

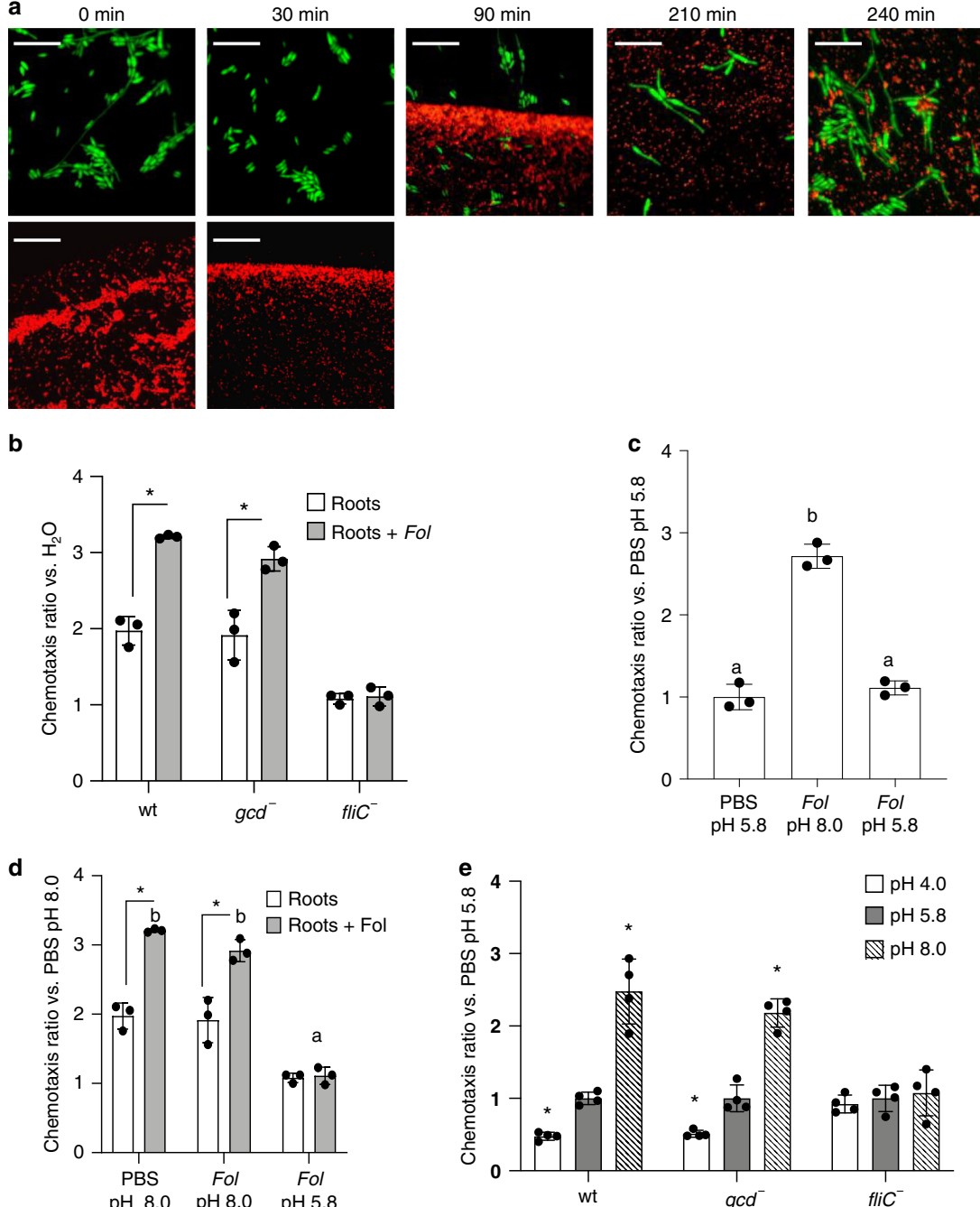

**Fig. 3 R. aquatilis exhibits chemotaxis toward F. oxysporum-derived alkaline pH gradients. a** RFP-tagged Ra36 and GFP-tagged Fol were spot-inoculated at 5 mm distance on soft agarose-coated slides and imaged at the indicated time points. Number of independent experiments ($n_{i.ex.}$) = 3, with two replicates each. Images taken at times 0 and 30 min show physically separated areas. Scale bar, 50 μm. **b–e** Capillary chemotaxis assay of the indicated Ra36 strains maintained in water **b** or phosphate-buffered saline (PBS) adjusted to pH 5.8 **c**, **e**, or 8.0 **d**, toward exudates from uninoculated (Roots) or Fol-inoculated tomato roots (Roots+Fol) **b**; exudates from Fol mycelium, either unadjusted (Fol 8.0) or adjusted to pH 5.8 (Fol 5.8) **c**, **d**; or PBS adjusted to the indicated pH values **e**. In **b–d** $n_{i.ex.}$ = 3; 2 technical replicates each, in **e** $n_{i.ex.}$ = 4; two technical replicates each. **b** *$P < 0.05$ versus tomato roots according to two-tailed, unpaired Student's $t$ test. **c**, **d** columns with the same letter are not significantly different according to one-way ANOVA followed by Tukey's multiple comparison test ($P < 0.05$). **e** *$P < 0.05$ versus PBS pH 5.8 according to one-way ANOVA followed with a Dunnett's post hoc test. Data are presented as mean values ± SD from three **b–d** and four **e** independent experiments. Source data from **b–e** are provided as a Source Data file.

preferential access to its ecological niche in the host plant. By harnessing the chemotropic ability of Fol hyphae, Ra36 cells significantly increase their chance of reaching and colonizing plant roots. Initial chemotaxis of Ra36 toward Fol is triggered by an alkaline pH gradient originating from the fungal hyphae. Once the bacterial cells have reached the fungus, they navigate

along the hyphae and progressively accumulate at the hyphal tip as the fungus approaches the plant root. In the rhizosphere, Ra36 secretes significant amounts of GlcA to cause extracellular acidification, thereby reversing Fol-induced host alkalinization and effectively blocking fungal infection. Because host alkalinization is a general infection mechanism used by fungal

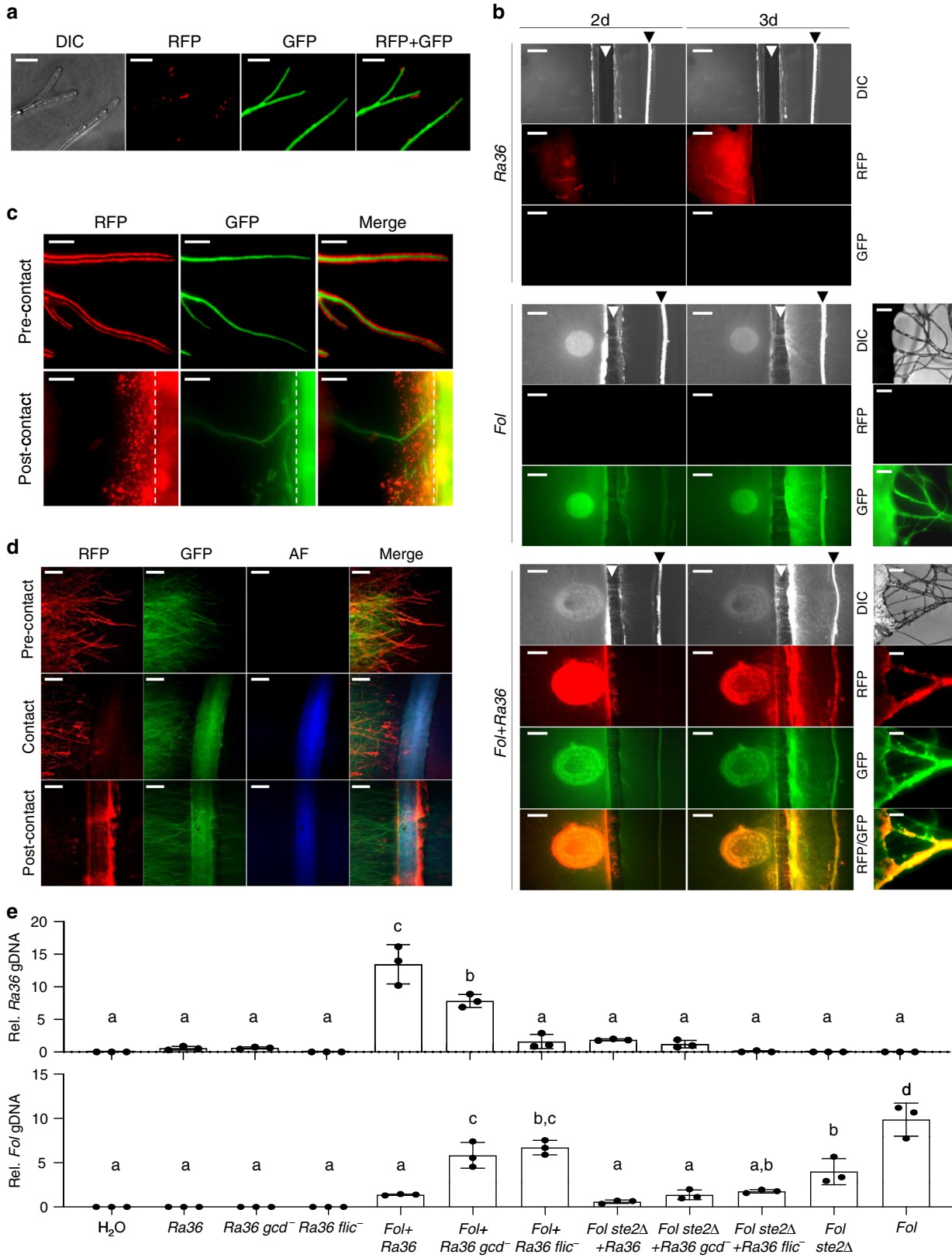

phytopathogens, and since acidification is widely conserved among soil bacteria[13,25,26], these findings could be of broad relevance for fungus–bacteria interaction in the rhizosphere as well as in other complex environments.

## Methods

**Fungal, yeast, and bacterial strains and culture conditions**. *Rahnella aquatilis* strain 36 (*Ra36*) was previously isolated from chickpea roots[23]. *Escherichia coli* and *Ra36* were routinely grown at 37 °C or 28 °C, respectively, in Luria Bertani (LB) medium (5 g/L yeast extract, 10 g/L peptone, 10 g/L NaCl, pH 7.0) containing

**Fig. 4 R. aquatilis exploits chemotropism of F. oxysporum hyphae to gain preferential access to tomato roots. a–d** Fluorescence microscopy showing RFP-tagged *Ra36* (RFP) and GFP-tagged *Fol* (GFP). DIC, differential interference contrast; AF, tomato root autofluorescence. **a** *Ra36* cells navigating along a film of liquid surrounding *Fol* hyphae and accumulating close to hyphal tips. Number of independent experiments ($n_{i.ex.}$) = 4, with two replicates each. Scale bar, 50 μm. **b** *Ra36* uses *Fol* hyphae to cross an air-filled space. *Ra36* and *Fol* were point-inoculated either individually or together on a soft-agar plate to the left of a medium-free gap (white arrowhead) and a tomato root was placed on the opposite side (black arrowhead). Representative images from plates after 2 and 3 days are shown. Magnified micrographs on the right show bundles of aerial *Fol* hyphae crossing the gap. $n_{i.ex.}$ = 4, with two replicates each. Scale bar, 2 mm (left) or 50 μm (right). **c, d** Representative micrographs showing *Fol* hyphae colonized by *Ra36*, before and after contact with a tomato root (experimental set-up shown in **b**. Note the accumulation of *Ra36* cells on the leading hyphae during the pre-contact stage, followed by rapid relocation to the plant root after contact. $n_{i.ex.}$ = 4, with two replicates each. Scale bar, 100 μm. **e** Colonization of tomato roots by *Ra36* is promoted by flagella-driven movement along chemotropic *Fol* hyphae. Tomato seedlings were planted in horticultural soil mixed or not with microconidia of the indicated *Fol* strain. After 3 days, the indicated *Ra36* strain or water (H$_2$O) was added at 5 cm distance from the plant hypocotyl. After 10 days, the relative amounts of bacterial (upper graph) and fungal (lower graph) DNA in the roots were measured by real-time qPCR. $n_{i.ex.}$ = 3; five plants/treatment. Data are expressed as ng of target DNA per 100 ng of plant DNA. Data are presented as mean values ± SEM from three independent experiments. Columns with the same letter are not significantly different according to one-way ANOVA followed by Tukey's multiple comparison test (*P* < 0.05). Source data from **e** are provided as a Source Data file.

kanamycin sulfate (50 μg/mL), ampicillin (250 μg/mL) or carbenicillin (100 μg/mL), where appropriate. *Saccharomyces cerevisiae* was grown in yeast extract peptone dextrose (YPD; 1% Bacto yeast extract, 2% bacto peptone, and 2% dextrose), except for the selection of recombinants which was carried out in yeast nitrogen base medium (Sigma-Aldrich, Madrid, Spain) containing the appropriate amino-acid supplements.

*F. oxysporum* f. sp. *lycopersici* race 2 isolate 4287 (FGSC9935; *Fol*) or previously reported *Fol* mutants either lacking the *ste2* gene[6] or constitutively expressing the GFP[7] were used throughout the work. For microconidia production, cultures were grown in potato dextrose broth (PDB) at 28 °C with shaking at 170 rpm[38]. For phenotypic analysis of *Fol* colony growth, serial dilutions (10$^6$, 10$^5$, 10$^4$, and 10$^3$ mL$^{-1}$) of freshly obtained microconidia were spotted onto YPD agar plates supplemented or not with 0.8% (w/v) GlcA (Sigma-Aldrich), incubated at 28 °C for 3 d and imaged. Experiments included three replicates and were performed at least three times with similar results. Microconidial and bacterial cell suspensions were routinely stored at −80 °C with 30% glycerol.

**Bioinformatic analysis**. Identification of *gcd* and *fliC* gene orthologs from different *R. aquatilis* strains was performed with the BLAST algorithm[39], using the *R. aquatilis* HX2 *gcd* and *fliC* genes as baits. Nucleotide sequences encompassing the entire *gcd* and *fliC* genes and comprising 1.5-kb upstream and downstream of the retrieved ORFs were aligned using the BioEdit 7.0.0 software (Ibis Bioscience, Carlsbad, CA, USA). Primer pairs Gcd7/Gcd8 and Flic7/Flic8 were designed on conserved upstream and downstream gene regions and used for amplification and sequencing of the two genes from *Ra36*. A complete list of the primer sequences used in this study is provided in Supplementary Table 1.

**Nucleic acid manipulations**. Isolation of genomic and plasmid DNA from yeast and bacterial cells was performed with the Puragene Yeast/Bact. Kit B and QIAprep Spin Miniprep Kit (Qiagen, Hilden, Germany) according to the manufacturer's instructions. Total genomic DNA from tomato seedlings was extracted by using the DNeasy Plant Mini Kit (Qiagen) according to the manufacturer's instructions. Routine nucleic acid manipulations were performed according to standard protocols[40].

**Rahnella aquatilis mutant generation**. The *Ra36 gcd$^-$* and *fliC$^-$* mutants were generated by targeted replacement with the kanamycin resistance marker. DNA fragments flanking the *gcd* and *fliC* coding regions were amplified from gDNA of *Ra36* using primer pairs Gcd1+Gcd2, Gcd3+Gcd4 and Flic1+Flic2, Flic3+Flic4, respectively. The kanamycin resistance marker was amplified from the pET28-c (+) plasmid (Novagen, Merck, Darmstadt, Germany) using primers Kanf+Kanr. The *S. cerevisiae* recombination cloning method[41] was used to assemble the complete deletion cassette in the pRS246 vector[42] linearized with *EcoRI/Hind*III, by co-transformation with 1 μg of each of the four DNA fragments into the yeast strain FY834[43]. The resulting plasmids pRS426.1 and pRS426.2 were isolated from yeast and cloned and propagated in the *E. coli* TOP10 strain (Invitrogen, Barcelona, Spain) using standard protocols[40]. The *gcd$^-$* and *fliC$^-$* deletion alleles were amplified from pRS426.1 and pRS426.2, respectively, with primers PRS426f +PRS426r and used to transform cells of the *Ra36* wild-type strain. Kanamycin-resistant transformants showing homologous insertion in the *gcd* and *fliC* coding regions were identified by PCR of gDNA with primers Gcd5+Gcd6 and Flic5 +Flic6, respectively. Allelic replacement of the *gcd* and *fliC* genes was further confirmed by DNA sequencing of a PCR fragment amplified from gDNA of the transformants with primers Gcd7+Gcd8 and Flic7+Flic8, respectively.

For fluorescence microscopy analysis, a red fluorescent strain of *Ra36* was obtained by transforming bacterial cells with 5 μg of plasmid FPB-31-444 (ATUM Bio, Newark, CA, USA) carrying the red fluorescent protein Rudolph-RFP and an

ampicillin resistance cassette, using a Bio-Rad Gene Pulser electroporator (Bio-Rad, Madrid, Spain) set at 1.8 kV, 200 W and 25 μF and 0.1 cm cuvettes.

**Extracellular pH determination and GlcA quantification**. For visual evaluation of *Ra36* acidifying activity, 5 μL drops containing 2.5 × 10$^6$ *Ra36* or *Ra36 gcd$^-$* colony forming unit (CFU) were spot-inoculated onto plates containing minimal medium with urea (MMU; 15 g L$^{-1}$ glucose, 0.5 g L$^{-1}$ MgSO$_4$·7(H$_2$O), 0.5 g L$^{-1}$ KCl, 1.0 g L$^{-1}$ KH$_2$PO$_4$, 1.5 g L$^{-1}$ urea, 20 g L$^{-1}$ agar) adjusted to pH 7. After 2 days of incubation at 28 °C, plates were overlaid with a 0.833 μM solution of the pH indicator bromocresol green or purple and imaged. Experiments were performed at least four times, with two replicates each.

To assess a possible inhibitory effect of *Ra36* on *Fol* colony growth and to visualize *Ra36*-mediated alkalinisation of *Fol* colony margins, 5 μL drops containing 5 × 10$^4$ *Fol* microconidia and 2.5 × 10$^6$ CFU mL$^{-1}$ *Ra36* or *Ra36 gcd$^-$* were spot-inoculated at a distance of 30 mm on MMU plates adjusted to pH 7.0. Visual analysis of pH change was performed by supplementing media with bromocresol purple (0.833 μM). Plates were incubated at 28 °C for 4–7 d before imaging. All fungus–bacteria confrontation assays were performed in three or more independent experiments with three replicates each.

For plate visualization of rhizosphere pH dynamics[12] during the *Fol-Ra36*–plant interaction, 2-week-old tomato seedlings (*Solanum lycopersicum* cv. Moneymaker; used throughout the study) were either left untreated or inoculated by dipping roots into a suspension of *Ra36* or *Ra36 gcd$^-$* cells in water (1 × 10$^9$ CFU mL$^{-1}$). Seedlings were then transferred onto 0.5% water agar plates supplemented with bromocresol purple (0.833 μM) and containing either *Fol* microconidia (1.25 × 10$^6$ mL$^{-1}$) or water (negative control). Plates were incubated at 25 °C with 14 h light and 10 h dark up to 3 d before imaging. Experiments included three replicate plates and were performed at least three times with similar results.

To record pH dynamics in liquid media, *Fol* microconidia (5 × 10$^5$ conidia mL$^{-1}$) were pre-germinated 14 h in MMU medium with shaking (170 rpm) at 28 °C, washed with sterile ddH$_2$O and shifted to fresh MMU medium adjusted to pH 7. Then, 5 × 10$^6$ CFU mL$^{-1}$ *Ra36* or *Ra36 gcd$^-$* were added and cultures were incubated at 28 °C and 170 rpm. Each hour, aliquots of culture supernatant were withdrawn, and pH was measured in a pH meter. Cultures containing either *Fol* or *Ra36* alone were used as controls. Experiments were performed on three to four independent occasions with three replicates each.

To monitor pH dynamics in liquid media in the presence of the plant, 5 × 10$^5$ mL$^{-1}$ *Fol* microconidia and 5 × 10$^6$ CFU mL$^{-1}$ *Ra36* or *Ra36 gcd$^-$* were inoculated in sterile H$_2$O. Then, a 2-week-old tomato plant was added to keep only the root submerged in the liquid. The culture was incubated, aliquots of culture supernatant were collected, and pH was measured as described above. Cultures containing either *Fol*, *Ra36*, or tomato plants alone were used as controls. Experiments were performed on three to four independent occasions with three replicates each.

To investigate rhizosphere pH dynamics in soil, 3-week-old tomato seedlings were dip-inoculated with a suspension of *Ra36* or *Ra36 gcd$^-$* cells (1 × 10$^9$ CFU mL$^{-1}$), transferred to non-sterilized or sterilized (autoclaved 20 min at 120 °C) horticultural soil obtained from a tomato field located in Riccia, Molise (Italy) and maintained in a growth chamber (14/10 h light/dark cycle) at 28 °C. After 2 weeks, 10 plants per treatment were collected, roots were cleaned from the soil, submerged in 5 mL sterile ddH$_2$O for 12 hours at 28 °C and extracellular pH was measured in a pH meter. Uninoculated plants were used as controls. Experiments were performed on four independent occasions with three replicates each.

D-GlcA concentrations in culture supernatants or tomato root exudates described above was determined using the K-GATE detection kit (Megazyme International, Bray, Ireland) according to the manufacturer's instructions. For all experiments, GlcA measurements were performed on three to four independent occasions with at least three replicates per treatment.

**Virulence-related assays**. Assays for the evaluation of *Fol* invasive growth on living plant tissue[7] were performed using tomato fruits (cultivar Rovente). In brief, the epidermis of surface sterilized tomato fruits was punctured with a pipette tip and 10 μL of bacterial suspension ($10^9$ CFU mL$^{-1}$) or water (control) was injected into the fruit tissue. After 30 min, 10 μL of a suspension of *Fol* microconidia ($5 \times 10^8$ mL$^{-1}$) or water (control) was injected into the same inoculation site. Fruits were incubated at 28 °C in a humid chamber and infection sites were imaged 5 days after inoculation (DAI). Experiments were performed on three independent occasions with three replicates each.

Cellophane invasion assays[9] were performed on MMU plates. In brief, 100 μL of a *Ra36* cell suspension (OD$_{600}$ of 0.1) or water (control) was uniformly spread on the culture medium before placing an autoclaved cellophane sheet on the plate (colorless; Manipulados Margok, Zizurkil, Spain). Next, $2 \times 10^5$ *Fol* microconidia were spot-inoculated at the center of the plate. After 4 days at 28 °C, plates were imaged (before), then the cellophane membrane was carefully removed, and plates were incubated for one additional day at 28 °C and imaged (after). MMU was either unbuffered or buffered to pH 5 or 7 with 100 mM 2-(N-morpholino) ethanesulfonic acid (MES). Experiments included three replicate plates and were performed three times with similar results.

For analysis of hyphal aggregation, fungal ($2.5 \times 10^6$ conidia mL$^{-1}$) and bacterial ($10^8$ CFU mL$^{-1}$) strains were co-cultivated for 48 h in liquid MMU at 28 °C and 170 rpm. Experiments included four replicates and were performed at least three times with similar results.

For root adhesion assays[7], entire roots of 2-week-old tomato seedlings were dip-inoculated with *Ra36* as described above, placed in Erlenmeyer flasks containing a suspension of $10^7$ mL$^{-1}$ *Fol* microconidia and incubated 3 days at 28 °C and 120 rpm. Fungal aggregates in liquid medium or on tomato roots were imaged in a Leica binocular microscope (Leica Microsistemas S.L.U., Barcelona, Spain) using a Leica DC 300 F digital camera. Experiments included four replicates and were performed at least three times with similar results.

**Plant infection assays**. Roots of 2-week-old tomato seedlings were dipped for 30 min into a suspension of *Fol* microconidia in water ($5 \times 10^6$ conidia mL$^{-1}$). For co-inoculation assays, plant roots were immersed for 2 h in a suspension of *Ra36* cells ($1 \times 10^9$ CFU mL$^{-1}$) and then dip-inoculated with *Fol* microconidia ($5 \times 10^6$ conidia mL$^{-1}$). To test the effect of inoculum density, suspensions of *Ra36* cells at different concentrations ($1 \times 10^{10}$; $1 \times 10^9$; $1 \times 10^8$; $1 \times 10^7$, and $1 \times 10^6$ CFU mL$^{-1}$) were used. After root inoculation, tomato seedlings were planted in vermiculite or in non-sterilized or sterilized horticultural soil (see above), maintained in a growth chamber (14/10 h light/dark cycle) at 28 °C and irrigated either with unbuffered water or with a solution of 1 mM MES buffer adjusted to pH 5.0 or pH 7.0.

Soil inoculation was performed by mixing non-sterilized or sterilized horticultural soil with *Ra36* cells and/or *Fol* microconidia at concentrations of $1 \times 10^8$ CFU and $4 \times 10^5$ conidia per g of soil, respectively.

Plant survival was recorded daily up to 55 days, calculated by the Kaplan–Meier method, and compared among groups using the log-rank test. All infection assays included fifteen plants per treatment and were performed at least three times with similar results. Data were plotted using the GraphPad Prism 5 software (GraphPad Software, La Jolla, CA, USA).

**Detection of *Fol* and *Ra36* in tomato plant tissue**. Qualitative assessment of fungal and bacterial colonization of tomato plants was performed 2 weeks after dip-inoculation of 2-week-old tomato roots with a suspension of *Ra36* cells ($1 \times 10^9$ CFU mL$^{-1}$) and/or *Fol* microconidia in water ($5 \times 10^6$ conidia mL$^{-1}$) as described above. Seedlings were removed from the vermiculite and the roots and stems were surface sterilized by submerging them into 2% sodium hypochlorite for 2 min. Plant tissues were rinsed twice with sterile distilled water and cut into 1 cm long sections, which were transferred onto MMU plates with or without 0.833 μM of the pH indicator bromocresol purple. Fungal or bacterial growth was recorded after 5 days of incubation at 28 °C. Experiments included three replicate plates per treatment and were performed at least three times with similar results.

Quantification of fungal and bacterial biomass in *Fol*- and/or *Ra36*-inoculated tomato plants grown in vermiculite was carried out by real-time qPCR[19], using total gDNA extracted from tomato roots 15 days after dip-inoculation with a suspension of *Ra36* cells ($1 \times 10^9$ CFU mL$^{-1}$) and/or *Fol* microconidia ($5 \times 10^6$ conidia mL$^{-1}$). To measure the effect of *Fol* in soil on the colonization ability of *Ra36*, 2-week-old tomato seedlings were planted in horticultural soil containing $2 \times 10^4$ microconidia g$^{-1}$ soil. After 3 days, 50 μl of a suspension of *Ra36* cells containing $8 \times 10^8$ CFU mL$^{-1}$ were added at a distance of 5 cm from the hypocotyl. After 10 days, total gDNA was extracted from the tomato roots and biomass of *Ra36* was measured by real-time qPCR.

Real-time qPCR reactions were performed with the SYBR® Premix Ex Taq™ (Takara Bio, Inc., Otsu, Japan) in a Eppendorf Mastercycler ep gradient S system (Eppendorf, Milan, Italy), using the primer pairs Gcd9 + Gcd10 (located outside of the deleted region of the *Ra36 gcd* gene), ACT2 + ACTQ6 (*Fol actin* gene) and GADPH1 + GADPH2 (tomato *gadph* gene) (Supplementary Table 1). Relative amounts of fungal and bacterial genomic DNA were calculated by comparative ΔΔCt with the tomato *gadph* gene. DNA concentrations in each sample were extrapolated from standard curves obtained by plotting the logarithm of known concentrations (10-fold dilution series from 10 ng to 1 pg/25 μL reaction) of *Fol*

and *Ra36* gDNA against the Ct values. To normalize the serially diluted DNA samples, 100 ng gDNA from non-inoculated plants was added to each sample of the dilution series. Real-time qPCR data represent the mean ± SE from three independent experiments, each with five plants per treatment.

**Collection of exudates from tomato roots and *Fol* hyphae**. To obtain tomato root exudate[6], uninoculated or *Ra36*-inoculated tomato roots ($1 \times 10^9$ CFU/mL) were placed in sterile ddH$_2$O in the absence or presence of $5 \times 10^5$ microconidia mL$^{-1}$ of *Fol*. After 48 h at 25 °C, the supernatant was sterilized by filtration through a 0.22-μm membrane (Merck Millipore) and stored at −20 °C until use.

To obtain *Fol* hyphal exudate[4], $1 \times 10^7$ microconidia mL$^{-1}$ were pre-germinated for 16 h in 50 mL diluted PDB (1:50; v:v in H$_2$O) at 28 °C with shaking at 170 rpm. Germlings were washed twice with sterile ddH$_2$O and incubated for 48 h in 5 mL sterile ddH$_2$O at 28 °C and 170 rpm. The supernatant was sterilized by filtration through a 0.22-μm membrane (Merck Millipore) and stored at −20 °C until use. When required, the pH was adjusted to 5.8 with 0.1 N HCl.

**Assays for fungal chemotropism and bacterial chemotaxis**. Plate preparation, chemoattractant application, and scoring of *Fol* germ tube redirecting toward gradients of root exudates obtained from untreated or *Ra36*-inoculated plants were performed by using a hyphal chemotropism assay[6]. In brief, $2.5 \times 10^6$ *Fol* microconidia mL$^{-1}$ were embedded in 4 ml water agar (WA; 0.5%, w/v) (Oxoid) and poured into a 9 cm Petri dish. Then, two parallel wells, each at 5 mm distance from the scoring line, were filled with 40 μL of the test compound or the solvent control solution. Scoring was done on five independent batches of cells ($n = 100$ cells per batch) for each test compound. Experiments were performed at least three times with similar results.

Chemotaxis capillary assays[32] were carried out as follows. *Ra36* wild-type, *Ra gcd$^-$*, and *Ra fliC$^-$* strains were grown in LB overnight at 28 °C, washed either with sterile ddH$_2$O, PBS, tomato root exudate or *Fol* hyphal exudate depending on the experiment, and diluted in the same medium to an OD$_{600}$ of 0.1. Aliquots of 250 μL bacterial suspension were added to individual wells of a 96-well microtiter plate together with a 10-μL capillary containing the test compound or the solvent control. For competing gradient assays, two capillaries containing the different test compounds were added to the well. Plates were incubated for 60 min at 28 °C, capillaries were carefully lifted, the content was serially diluted and plated onto LA medium, and CFUs were counted 48 h after incubation at 28 °C. The following chemoattractant compounds and concentrations were tested: glutamine (Gln), tryptophan (Trp), all at 295 mM; glucose (Gluc), galactose (Gal), all at 50 mM. The chemotaxis ratio was calculated by dividing the number of bacteria in the tube containing the test compound by the number of bacteria in the tube containing the solvent control. All experiments included two replicates and were performed at least three times with similar results.

Flagellum-dependent swimming motility assays[44] were performed as follows. In brief, LB plates (0.3% w/v agarose) were spot-inoculated in the center with 5 μl of an overnight culture of the *Ra36* wild-type or *fliC$^-$* strain, incubated 24 h at 30 °C and colony radial growth was imaged. Experiments were performed three times, with four replicates each.

**Fluorescence microscopy**. For microscopic observation in tomato roots of GFP-tagged *Fol* and RFP-tagged *Ra36* strains, 2-week-old tomato seedlings were dip-inoculated into suspensions of *Fol* microconidia ($5 \times 10^6$ conidia mL$^{-1}$) and/or *Ra36* cells ($1 \times 10^9$ CFU mL$^{-1}$), planted in moist vermiculite and maintained in a growth chamber (14/10 h light/dark photoperiod) at 28 °C. Two and four DAI, roots were gently washed to remove the adhering vermiculite, incubated in a 95% per-fluorodecalin solution (Sigma-Aldrich), stained for 2 min with 0.005% (w/v) cal-cofluor white (CFW; Sigma-Aldrich) to visualize the plant cell wall and imaged. Experiments were performed three times, with two replicates each.

To observe *Ra36* swarming toward *Fol* hyphae, 2 μL drops containing $5 \times 10^4$ *Fol-GFP* microconidia or $2.5 \times 10^6$ CFU *Ra36-RFP* were spot-inoculated at a distance of 5 mm on a $1 \times 1$ cm square pad of soft (0.25% w/v) agarose MMU medium placed on top of a microscope glass slide. Images were recorded every 30 min up to 4 h post inoculation. Experiments were performed three times, with two replicates each.

To study bacterial movement along fungal hyphae, 2 μL drops containing $5 \times 10^4$ *Fol-GFP* microconidia or $2.5 \times 10^6$ *Ra36-RFP* CFU were spot-inoculated at a distance of 30 mm on MMU medium plates. Microscopic observation of *Fol* hyphae was performed 48 h after fungal and bacterial colonies had merged. Experiments were performed four times, with two replicates each.

To determine the movement of *Ra36* and *Fol* hyphae across a medium-free space, $5 \times 10^4$ *Fol-GFP* microconidia and $2.5 \times 10^6$ *Ra36-RFP* CFU were spot-inoculated either individually or together on soft (0.7% w/v) agarose MMU plates. Next, a 2-day-old tomato seedling was placed at a distance of 15 mm, and a 5-mm wide medium-free gap was created between the inoculation point and the tomato root by removing the medium with a sterile spatula. Microscopic observation was performed daily up to 3 DAI to follow *Ra36* and *Fol* dynamics over time. To qualitatively assess the presence of *Ra36* and *Fol* on the plant roots, tomato seedlings were gently removed from the plate at 2 DAI, rinsed under sterile H$_2$O

and stained with CFW, as described above. Experiments were performed at least three times on two or more separate days.

Low-resolution imaging was performed using a SteREO Lumar.V12 fluorescence stereomicroscope (Zeiss, Barcelona, Spain). Wide-field fluorescence imaging was performed using a Zeiss Axio Imager M2 Dual Cam microscope (Zeiss) equipped with a Photometrics Evolve EM512 digital camera (Photometrics Technology, Tucson, AZ, USA). Examination using epifluorescence (×400 magnification) was performed with the following filter blocks: CFW staining (G 365, FT 395, LP 420), RFP (BP 560/40, FT 585, BP 630/75), GFP (BP 450/490, FT 510, LP 515). Images were captured and processed using Axiovision 4.8, ZEN lite 2.3 (both from Zeiss) or ImageJ (v1.52)[45].

**Statistical analysis**. Percentage of plant survival was compared among treatment groups using the log-rank test. For multiple-group comparisons, a one-way analysis of variance was used for testing no differences among the group means. Post hoc comparisons were adjusted using Dunnett's or Tukey's corrections (chemotaxis and real-time qPCR data). Comparisons between two groups were carried out using a two-tailed unpaired Student's $t$ test (chemotaxis data). A Yates' corrected Chi-squared test (two-sided) was used to determine significant differences between the observed frequencies of fungal germ tubes pointing toward the chemoattractant or the solvent control (chemotropism data).

Real-time qPCR data are presented as mean ± SE. Data from chemotaxis and chemotropism experiments and from pH and GlcA measurements are presented as mean ± SD.

Statistical analyses were performed using GraphPad Prism 5 software (GraphPad Software). In all cases a value of $P < 0.05$ was considered statistically significant.

**Reporting summary**. Further information on research design is available in the Nature Research Reporting Summary linked to this article.

## Data availability

The *R. aquatilis* HX2 *gcd* and *fliC* genes (GenBank A.N.: "EF090904" and "AFE58723", respectively) were used to design primers for the amplification of the *gcd* and *fliC* gene orthologs from different *R. aquatilis* strains. Sequence data are deposited in the European Nucleotide Archive under accession number "LT708303" and in Genbank under accession number "MN972473", respectively. The statistical tests used to analyze each experimental data set as well as all the obtained significances and exact *P* values are provided in the Supplementary data 1 file. All other data are available from the corresponding authors upon reasonable request. Source data are provided with this paper.

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

## Acknowledgements

We thank the members of the Di Pietro laboratory for helpful comments on the manuscript. This work was supported by grants BIO2016-78923-R from the Spanish Ministerio de Economía y Competitividad (MINECO) to A.D.P. and PRIN-20089LSZ2A_003 from the Italian Ministero dell'Istruzione, dell'Università e della Ricerca (MIUR) to G.L. S.V. was supported by Marie Curie ITN FUNGIBRAIN (FP7-PEOPLE-ITN-607963).

## Author contributions

D.P., S.V., G.L., A.D.P., and D.T. designed the experiments. D.P., S.V., and D.T. carried out the experiments. D.P., S.V., G.L., A.D.P., and D.T. analyzed the data. D.P., A.D.P., and D.T. wrote the manuscript.

## Competing interests

The authors declare no competing interests.
