## [Peer Review File · Nature Communications]

Reviewers' comments:

Reviewer #1 (Remarks to the Author):

This paper characterizes interactions between the fungal pathogen *Fusarium oxysporum* (Fo) and the bacterium *Rahnella aquatilis* (Ra). Fo inhibits plant growth. Previous work demonstrated that Ra can prevent damage to plants by Fo, but the specific mechanisms by which this occurs have not been identified. In this manuscript, the authors demonstrate that Ra takes advantage of the migration of Fo to the plant host by hitching a ride on hyphae. The ability of bacteria to colonize and migrate along fungal hyphae is well known and has been demonstrated in many systems. The new aspects of the work presented in this manuscript are showing potential mechanisms of interactions between Fo and Ra. More specifically, the authors identify acidification of the environment by Ra as a mechanism by which it inhibits Fo growth and prevents damage to the host plant.

One of my main concerns with this work is understanding if and how this interaction actually occurs on plants growing in soil. Almost all of the work presented in the manuscript is done with plants, fungi, and bacteria growing in Petri dishes on agar based or liquid media. The only experiment to use soil (lines 134-141) is where Ra was added to plants grown in soil with or without Fo. This work only shows how Ra increases in abundance with Fo and does not measure the chemical or physical aspects of the interaction. All other work showing how the fungus and bacterium interact was done on media without other microbes present. This is a useful place to start to characterize this interaction, but completely removes the reality and ecological context from this interaction. How does the presence of other bacteria or fungi (or a complete soil bacterial/fungal community) impact this interaction? Does Ra actually cause acidification of soil through the production of gluconic acid or is this only really observed in agar or liquid media? These questions need to be answered to demonstrate the relevance of the reduced Petri dish conditions to soil systems where plants, Ra, and Fo would be interacting.

Another issue with the proposed work is the focus only on secretion of gluconic acid as the mechanism driving this interaction. Is this the only driver of the interaction dynamics? Or was it just the easiest to chase in this work? It makes sense and I understand why the authors explored this given the importance of alkalization by Fo to infect plants. But an unbiased screen (such as a mutagenesis screen of the bacterium) would have more systematically identified the major drivers of the interaction. For example, what is the relative importance of motility vs. gluconic acid production in driving this bacterial-fungal interaction? A knockout of flagella in Ra (or other drivers of active motility) would demonstrate that the spread of Ra on Fo via motility is essential for it to colonize plants.

The authors do not present data demonstrating the robustness of this interaction across a variety of conditions. For example, how sensitive is the interaction to initial levels of bacterial and fungal inoculum? It seems like too much of Ra initially could lead to inhibition of the fungus and then lack of hyphae to gain access to the plant. There are no experiments that I could find that tested relative amounts of bacteria and fungi to understand how initial concentrations impact interaction outcomes. How did the authors decide on the amounts that were used (lines 194-198). Do other genotypes of the fungus and the bacterium demonstrate the same interaction or is it limited to just the two presented? Does this interaction occur across other hosts (besides tomato)? Moisture can dramatically impact the ability of bacteria to spread on fungal hyphae and there are no experiments demonstrating how water availability impacts the outcomes of this interaction. These questions are not resolved and need to be in order to show robustness and generalizability of the interaction and proposed mechanism of interaction.

I am a big fan of conceptual figures in papers to help summarize major findings for the reader. But I am not sure Figure 4 is really helpful. It doesn't connect disparate experiments. It doesn't really make any confusing concepts more clear. It is really just a cartoon of what we can see in Figure 3.

I suggest that the authors make the figure have greater utility or remove it.

There are some really striking and helpful images/datasets that are currently tucked in Extended Data that would be helpful in the main figures. For example, I found Extended Data Figure 2d to be helpful.

Reviewer #2 (Remarks to the Author):

The manuscript of Palmieri et al shows a very nice study on the interaction of the plant colonizing bacterium *Rahnella aquatilis* 36 and the plant pathogenic fungus *Fusarium oxysporum* f. sp. *lycopersici*. The authors have performed a large, well designed and comprehensive study revealing how a plant beneficial bacteria shows pH-mediated chemotaxis towards a root-attacking pathogen, how it uses the fungal hyphae as means of transport to reach plant roots, and how it prevents pathogen attack by acidification of its surroundings thereby inhibiting early pathogenicity traits requiring a high pH. They show, using a mutational approach, that the mechanism responsible for this acidification is the production of gluconic acid. Although the fact that soil/rhizobacteria can hijack fungal hyphae for transport is not new, this study goes far beyond that and reveals new and exciting insights into biocontrol bacterium-fungal pathogen interactions such as pH mediated chemotaxis and pH-mediated biocontrol activity. The study is illustrated with a large set of informative and very nice microscopy pictures showing in detail the transport of the bacterium by the hyphae and the prevention of *Fusarium* growth inside the root vessels in the presence of the bacterium. The story is well written and told and of high scientific quality. However, I see some critical issues preventing the publication of this manuscript in its present form.

General comments:

I suggest to change the title of the manuscript because the highlight of this paper is not the fact that Ra36 can use Fol as highway (this fact is not new and has been shown for several rhizosphere/soil bacteria and several fungi). The main point is the fine-tuned interaction between the two microorganisms e.g the chemotaxis of the bacteria towards Fol, the role of gluconic-acid mediated acidification resulting in impairing the fungal infection process.

There are two main weak points throughout the manuscript:

First, you write that every experiment was performed 3-4 times with similar results, however, you always show only one of these experiments. To my opinion it is absolutely necessary to show at least one of the repetitions in the extended data at least for the major experiments such as the plant survival (Fig 1a and 1g), qPCR assays for monitoring root colonization (Fig 1b), the acidification/gluconic acid production (Fig 1f) as well as for the chemotaxis exps shown in Fig 2b-e. You could also provide the significances of the experimental repetitions in the extended tables. I am convinced that to show at least one repetition with similar results is absolutely necessary to prove that the observed effects are constant.

Second, it is absolutely not clear what data sets the individual figures are based on and what the experimental set up was. It is never indicated how many replicates were made, which is absolutely essential. Please indicate in the material sections exactly how many independent experiments were performed and how many replicates per treatments were used in each of these independent experiments. Always describe in the figure legends what is exactly shown. You only indicate: n iex = 3. But how many replicates were used per independent experiment? Is what you show in the individual figures the results of only one of these 3 independent experiments? If yes, averages of how many replicates are shown? Please give details of the exp set ups in the figure legends and in the methods (see also specific comments for the figs below).

Specific comments:

In order to complete the line of evidences necessary for demonstrating the role of gluconic acid in acidification and biocontrol the gcd mutant would have to be complemented and the complemented mutant should have the same effect as the wildtype strain. Why was this control not included in the experiments?

Why were the qPCR assays for monitoring of root colonization and the plant infection assays not performed in the same experimental setup (rot col. in peat/perlite, plant infection in vermiculite)? It would have been much more informative. As you present your data now, we do not know the microbial colonization levels in the plant infection experiment what would be important. Why did you use vermiculite only for the plant infection assay? Why not peat?

Line 40: What do you mean by reference strain? Do you mean this is the type strain of the forma specialis *Lycopersici*?

Lines 142 ff. Your only evidence for the fungal hyphae being used as a vector by bacterium is the microscopic in vitro assay shown in Fig. 3. Whether this results in a faster and easier colonization of plant roots by the bacteria remains unknown. Similarly all chemotaxis, plant protection and acidification experiments were also performed in vitro (I regard vermiculite as a nearly-in vitro situation) The situation in nature might be completely different. Please consider this in your discussion. Did you ever investigate these interactions under more natural conditions? Did you ever perform the biocontrol experiment in soil? To my opinion an experiment is lacking investigating whether the bacterium reaches the plant earlier (e.g. if inoculated at a certain distance of the plant stem) and the root colonization is established earlier in presence of the fungus.

Material and method section: it is very hard to figure out which methods were used for which figure (subfigure). Please indicate this in the method section.

LL 210-22 and LL270-282 combine and make a separate aPCR chapter. Description of qPCR conditions are simply missing.

Line 481. I don't understand the colors described in the legend. As it is indicated here, the purple area is the soil, but it surely is not. Isn't it the liquid film surrounding the hyphae? Please also indicate the hyphae and the root in the picture. And what are the orange circles? Are these soil particles? In general, I don't think this picture is very informative and would omit it.

Figure 1: the letters of subfigs g and f should be exchanged to keep a an order from top to down and from left to right (1f should be the figure bottom left and 1g the figure bottom right).

Figure 2 and extended data Fig 3a and c: . You indicate that * means sign. different at $P < 0.0001$, but based on what test? Please add statistics.

Fig. 3. I am not sure, whether all these figs should be shown in the main text. e.g Fig 3c could be moved to the extended data.

Extended data Fig 2: echange letters h and g, it is confusing that g is the figure bottom right and h the one bottom left. 3g: Neither in the figure nor in the legend there is an indication which plates were treated with water and which with Fol conidia. Please add.

Comments on experimental setups:

Fig 1a: and 1g, extended Data Fig. 1i and 1k: How many seedlings per treatment were used? Show at least one repetition of the experiment in the extended data or the significances of the repetitions in extended data tables 1-4 (it would also be helpful for the reader to indicate in the extended tables to which Figs these statistics belong)

Fig 1b and 3e: As I can extract from the Figure legend, the data shown in this Figure are averages of three technical not biological replicates. In this case, it is absolutely necessary to show results of experimental repetitions, since in fact what you show here are data from only a single biological replicate per treatment.

Fig 1f: Is the methods described in lines 257-262 the one for this figure? What is shown in this Figure? Results of one experiment? What are these averages based on? How many replicates? Show one repetition of experiment in extended data.

Fig. 2 b-2e, see comment Fig 1f. What are the averages shown here based on? Are these averages of three independent experiments with only one replicate per experiment? Or averages of several replicates of one of the three experiments? If yes, how many replicates were used to create these averages?

Figure3 e: see comments on Fig 1b.

Extended data Fig 3a, d, e and f, see comments on Fig 1f, are results of 1 of the 3, respectively 4 independent expts shown? If yes how many replicates were used?

Reviewer #3 (Remarks to the Author):

Summary

Biocontrol activities by root-associated bacteria are so far mainly explained by their direct antimicrobial compounds to pathogens or their competitions with pathogens for nutrients or places. The work has represented a new mechanism by which a bacterial endophyte protect tomato plants from fungal destructive pathogen *F. oxysporum*.

The authors have shown that secretion of gluconic acid (GcIA) by the endophytic rhizobacterium *Rahnella aquatilis* inhibits *F. oxysporum*-induced alkalisation, which is a critical step for its virulence, and even induce acidification in rhizosphere. Consistent with this, the bacterial endophyte efficiently protects tomato plants from the pathogens in a dependent manner of GcIA secretion. Furthermore, the authors have shown that the root endophytic bacteria are attracted by *F.ox* hyphae possibly via sensing alkaline pH gradient and use the hyphae as a highway to reach tomato plants in soil.

Collectively, the work has tried to propose a new strategy for endophytic bacteria to reach host efficiently as well as new mechanisms by which bacteria endophytes exhibit biocontrol activities against pathogens.

Overall, the story is interesting with smooth logical flow. I enjoy reading this.

However, I would like to propose several experiments. Especially, it would be nice if the authors would provide more mechanistic insights about how secretion of GcIA lead to inhibit fungal colonization via inhibition of *F. oxysporum*-induced alkalisation and also how bacterial endophytes are attracted by fungal hyphae (See my comments below). Addressing majority of major points below would be necessary to meet the criteria of potential readers of this journal.

Major points

- It is still not clear how secretion of GcIA by bacterial endophytes overcomes fungal-mediated alkalization, despite the fact that fungal pathogens should also secrete peptides that induce alkalization at the same timing. Is this just because the root endophyte bacteria secrete too much GcIA (bacterial dose dependence)? Alternatively, does secretion of GcIA (and/or endophytes) also inhibit fungal processes that are necessary for alkalization (Masachis, S. et al. Nature Microbiology)?

- It is necessary to provide more detailed mechanisms by which bacteria are attracted by *F. ox* hyphae. For this purpose, It might be useful to test whether bacteria attraction to fungal hyphae (especially at the hyphal tip) is related with fungal-mediated alkalization. To test this possibility, using fungal mutants that cannot secrete alkalization peptides would be useful (Masachis, S. et al. Nature Microbiology). Related with this, it might be also useful to test whether bacteria movement to host roots through hyphae depend on secretion of GclA by using bacteria mutants without GclA secretion (*gcd-*). As the concept that bacteria use fungal hyphae as a highway have been reported already (e.g., Yuanchen Zhang et al., Nature communications 2018), I think it would necessary to provide more mechanistic details in some ways.

- It seems that only one *gcd-* mutant line is described in this work. It is necessary to test multiple mutants or the complementation lines to validate that defects in the gene is responsible for the phenotype shown in at least some of the figures.

- It would be also interesting to test whether bacteria flagella is necessary to efficiently move on fungal hyphae and to exhibit bacteria-mediated biocontrol activities. Although I predict that flagella is necessary to move efficiently on fungal hyphae, it might be not essential for the bacteria biocontrol activities when bacteria are directly inoculated onto tomato roots (like Fig.1).

Minor points

- The interactions shown here does not seem to provide any benefits to fungal-side. Do you think the type of interactions is really sustainable in natural ecosystems? Is this related with the fact that GclA treatment improves *F. ox* growth in vitro (Extended Fig 2h)?

Response to reviewer s:

We are grateful to the three expert reviewers for their positive comments, constructive criticisms and suggestions. In the revised version of the manuscript we have now addressed most of the points raised by the reviewers. This included a substantial amount of additional experimental work. We believe that these amendments have strongly improved the manuscript.

Below is a list of detailed point-to-point responses:

Reviewer #1 (Remarks to the Author):

R1: One of my main concerns with this work is understanding if and how this interaction actually occurs on plants growing in soil. Almost all of the work presented in the manuscript is done with plants, fungi, and bacteria growing in Petri dishes on agar based or liquid media. The only experiment to use soil (lines 134-141) is where Ra was added to plants grown in soil with or without Fo. This work only shows how Ra increases in abundance with Fo and does not measure the chemical or physical aspects of the interaction. All other work showing how the fungus and bacterium interact was done on media without other microbes present. This is a useful place to start to characterize this interaction, but completely removes the reality and ecological context from this interaction. How does the presence of other bacteria or fungi (or a complete soil bacterial/fungal community) impact this interaction? Does Ra actually cause acidification of soil through the production of gluconic acid or is this only really observed in agar or liquid media? These questions need to be answered to demonstrate the relevance of the reduced Petri dish conditions to soil systems where plants, Ra, and Fo would be interacting.

Response: We fully agree with the reviewer on the need to study the tri-partite interaction among Fol, Ra36 and the tomato plant in natural soil. We therefore repeated all the key experiments in both unsterilized and sterilized soil obtained from an agricultural setting, and these results are now included in the revised version of the manuscript. Experiments conducted in soil include the determination of plant mortality (Fig. 1f, Supplementary Figs. 3g and 4a-b) as well as measurements of pH and gluconic acid production (Fig. 2e). Importantly, the results of these experiments show that GlcA-mediated acidification of the plant rhizosphere and protection against Fol-induced mortality by Ra36 are also active in a natural soil, both in the presence and absence of the native microbial soil community. Additionally, to address the role of gluconic acid production we quantified bacterial and fungal biomass in tomato roots grown in soil, using the wild type Ra36 strain as well as the *gcd* deletion mutant (Fig. 4e). Importantly, we found that flagellar motility is essential for Ra36 to efficiently reach plant roots in the presence of Fol in the soil (Fig. 4e) and to suppress Fol-triggered mortality. Altogether these experiments demonstrate that disease suppression by Ra36 is also effective under ecologically relevant conditions such as those naturally found in agricultural soils.

R1: Another issue with the proposed work is the focus only on secretion of gluconic acid as the mechanism driving this interaction. Is this the only driver of the interaction dynamics? Or was it just the easiest to chase in this work? It makes sense and I understand why the authors explored this given the importance of alkalization by Fo to infect plants. But an unbiased screen (such as a mutagenesis screen of the bacterium) would have more systematically identified the major

drivers of the interaction. For example, what is the relative importance of motility vs. gluconic acid production in driving this bacterial-fungal interaction? A knockout of flagella in Ra (or other drivers of active motility) would demonstrate that the spread of Ra on Fo via motility is essential for it to colonize plants.

Response: We thank the reviewer for this excellent suggestion. While we believe that the focus on gluconic acid secretion is justified by the key role of pH in regulating virulence mechanisms of Fol, we agree that including a flagellar mutant of Ra36 is of interest to understand the relative contribution of bacterial motility. We therefore generated Ra36 mutants lacking the orthologue of the *fliC* gene, encoding a structural flagellin essential for flagellum function and bacterial motility (Blair 1995. *Annu. Rev. Microbiol.* 49:489–522). As expected, the *fliC*⁻ mutants failed to produce swimming zones on soft agar plates (Supplementary Fig. 2h) and failed to exhibit chemotaxis towards tomato roots or pH gradients, in contrast to the wild type Ra36 strain (Fig. 3b, e). Moreover, the ability of the *fliC*⁻ mutant to colonize tomato plants was drastically reduced compared to the Ra36 wild type strain, when inoculated in soil containing Fol at 5 cm distance from plant hypocotyl (Fig. 4e). In line with this, protection of tomato plants against Fol infection was independent of FliC-mediated bacterial movement when Ra36 was inoculated directly on the plant roots, but not when inoculation was performed at 5 cm distance (Supplementary Fig. 3g and 4b).

R1: The authors do not present data demonstrating the robustness of this interaction across a variety of conditions. For example, how sensitive is the interaction to initial levels of bacterial and fungal inoculum? It seems like too much of Ra initially could lead to inhibition of the fungus and then lack of hyphae to gain access to the plant. There are no experiments that I could find that tested relative amounts of bacteria and fungi to understand how initial concentrations impact interaction outcomes. How did the authors decide on the amounts that were used (lines 194-198). Do other genotypes of the fungus and the bacterium demonstrate the same interaction or is it limited to just the two presented? Does this interaction occur across other hosts (besides tomato)? Moisture can dramatically impact the ability of bacteria to spread on fungal hyphae and there are no experiments demonstrating how water availability impacts the outcomes of this interaction. These questions are not resolved and need to be in order to show robustness and generalizability of the interaction and proposed mechanism of interaction.

Response: We are grateful to the reviewer for this suggestion. Throughout this study we used a concentration of Fol (5×10^6 ml⁻¹), which was previously optimized in our group to provide robust and reproducible infection and disease symptoms on tomato plants (Lopez-Berges et al. 2010, *Plant Cell*). Therefore, to test the effect of the relative concentrations of bacterial versus fungal inoculum, we have now performed co-inoculation experiments, where we applied increasing concentrations of Ra36. The results clearly show that the protective effect of Ra36 against Fol is concentration dependent (Fig. 1a).

On the other hand, since the biocontrol activity of Ra36 isolates against different fungal pathogens (*F. oxysporum* f. sp. *ciceris*, *Penicillium expansum*, *Botrytis cinerea*) and across different plant hosts (chickpea, apple) was previously demonstrated (Palmieri et al. 2016, *Plant and Soil*; Calvo et al. 2007, *Int J Food Microbiol*), we opted against performing experiments with other plant pathogens or host plants in this study.

Regarding moisture, we agree with the reviewer that flagellated bacteria require high relative humidity in the soil to be able to freely move, disperse and eventually reach their hosts (Dechesne et al. 2010, Proc Natl Acad Sci U S A). Indeed, we found that Ra36 cells were unable to cross water unsaturated environments (a 5 mm air-filled space in our experimental setting) in the absence of fungal hyphae, but were able to do so in the presence of Fol (Fig. 4b). This suggests that the liquid film surrounding Fol hyphae provides a sufficient water-continuum to support movement of Ra36 across water-unsaturated soil zones. This is in agreement with an increasing body of literature indicating that fungal hyphae are responsible for the redistribution of both nutrients and water through the soil and that bacteria benefit from their association with fungi to overcome movement limitations in dry and oligotrophic environments (please see Furuno et al. 2010, Environ Microbiol; Martin et al. 2013, Appl Environ Microbiol; Guhr et al. 2015, Proc Natl Acad Sci U S A; Worrich et al. 2016 Appl Environ Microbiol; Worrich et al. 2017 Nat Commun).

R1: I am a big fan of conceptual figures in papers to help summarize major findings for the reader. But I am not sure Figure 4 is really helpful. It doesn't connect disparate experiments. It doesn't really make any confusing concepts more clear. It is really just a cartoon of what we can see in Figure 3. I suggest that the authors make the figure have greater utility or remove it.

Response: We fully agree with this appreciation and have removed the Figure.

R1: There are some really striking and helpful images/datasets that are currently tucked in Extended Data that would be helpful in the main figures. For example, I found Extended Data Figure 2d to be helpful.

Response: We agree with the reviewer. We have now completely restructured the content of the main and the supplementary figures, including transfer of the indicated panel to Fig. 2b.

Reviewer #2 (Remarks to the Author):

R2: The manuscript of Palmieri et al shows a very nice study on the interaction of the plant colonizing bacterium *Rahnella aquatilis* 36 and the plant pathogenic fungus *Fusarium oxysporum* f. sp. *lycopersici*. The authors have performed a large, well designed and comprehensive study revealing how a plant beneficial bacteria shows pH-mediated chemotaxis towards a root-attacking pathogen, how it uses the fungal hyphae as means of transport to reach plant roots, and how it prevents pathogen attack by acidification of its surroundings thereby inhibiting early pathogenicity traits requiring a high pH. They show, using a mutational approach, that the mechanism responsible for this acidification is the production of gluconic acid. Although the fact that soil/rhizobacteria can hijack fungal hyphae for transport is not new, this study goes far beyond that and reveals new and exciting insights into biocontrol bacterium-fungal pathogen interactions such as pH mediated chemotaxis and pH-mediated biocontrol activity. The study is illustrated with a large set of informative and very nice microscopy pictures showing in detail the transport of the bacterium by the hyphae and the prevention of *Fusarium*

growth inside the root vessels in the presence of the bacterium. The story is well written and told and of high scientific quality.

Response: We are grateful to the reviewer for the positive appreciation of our work.

R2: I suggest to change the title of the manuscript because the highlight of this paper is not the fact that Ra36 can use Fol as highway (this fact is not new and has been shown for several rhizosphere/soil bacteria and several fungi). The main point is the fine-tuned interaction between the two microorganisms e.g the chemotaxis of the bacteria towards Fol, the role of gluconic-acid mediated acidification resulting in impairing the fungal infection process.

Response: We have changed the title to "Bacterial endophyte exploits chemotropism of a fungal pathogen for plant colonization" to remove the highway concept and include the chemotropism. Due to the strict limitation in the number of characters in the title, we were unable to also include other details of the fine-tuned interaction such as the secretion of GlcA.

R2: There are two main weak points throughout the manuscript:

First, you write that every experiment was performed 3-4 times with similar results, however, you always show only one of these experiments. To my opinion it is absolutely necessary to show at least one of the repetitions in the extended data at least for the major experiments such as the plant survival (Fig 1a and 1g), qPCR assays for monitoring root colonization (Fig 1b), the acidification/gluconic acid production (Fig 1f) as well as for the chemotaxis exps shown in Fig 2b-e. You could also provide the significances of the experimental repetitions in the extended tables. I am convinced that to show at least one repetition with similar results is absolutely necessary to prove that the observed effects are constant.

Response: While it is true that for virulence assays only one of the 3 independent experiments were shown in the previous version of the manuscript, for all other types of experiments (qPCR, acidification/gluconic acid production, chemotaxis, chemotropism) the results shown in the previous graphs represented the mean \pm SD or SE from the different independent experiments (biological replicates). We have now clarified this concept in the figure legends and in the methods section. We agree with the point of the reviewer, who suggested to show at least one additional repetition and therefore now include in the revised manuscript the results of a second biological replicate for key plant survival experiments (Supplementary Fig. 4). We have also added a separate excel file (Supplementary Data 1) containing the statistical tests used to analyze each experimental dataset as well as all the obtained significances and P values. Moreover, following the Nature Journals policy, the raw data underlying all the reported mean values are provided as a Source Data file.

R2: Second, it is absolutely not clear what data sets the individual figures are based on and what the experimental set up was. It is never indicated how many replicates were made, which is absolutely essential. Please indicate in the material sections exactly how many independent experiments were performed and how many replicates per treatments were used in each of these independent experiments. Always describe in the figure legends what is exactly shown. You only indicate: $n_{\text{exp}} = 3$. But how many replicates were used per independent experiment? Is what you show in the individual figures the results of only one of these 3 independent experiments? If yes,

averages of how many replicates are shown? Please give details of the exp set ups in the figure legends and in the methods (see also specific comments for the figs below).

Response: We fully agree with the reviewer that these points were not sufficiently detailed in the first version of the manuscript. In the revised version, we now present a detailed description of the experimental set up as well as the number of independent experiments and replicates for each figure panel. This information has been included in the figure legends as well as in the corresponding Methods section.

R2: Specific comments:

In order to complete the line of evidences necessary for demonstrating the role of gluconic acid in acidification and biocontrol the *gcd* mutant would have to be complemented and the complemented mutant should have the same effect as the wildtype strain. Why was this control not included in the experiments?

This point was raised by reviewers #2 and #3. We agree with both reviewers on the need to test either complemented lines or multiple mutants of the *gcd* gene at least in the key experiments. In the revised version of the manuscript, we included the results for an additional independent knockout mutant, as well as for an ectopic transformant in which the integration of the gene knockout cassette occurred in a different genomic region. As expected, the second knockout mutant was also impaired in production of GlcA, extracellular acidification and control of Fol infection, whereas the ectopic transformant was similar to the Ra36 wild type strain (Fig. 2d and Supplementary Fig. 1a). An analogous approach was used to validate the effect of the deletion of the *fliC* gene on bacterial motility (Supplementary Fig. 2h).

R2: Why were the qPCR assays for monitoring of root colonization and the plant infection assays not performed in the same experimental setup (rot col. in peat/perlite, plant infection in vermiculite)? It would have been much more informative. As you present your data now, we do not know the microbial colonization levels in the plant infection experiment what would be important. Why did you use vermiculite only for the plant infection assay? Why not peat?

Response: We apologize for the misunderstanding on this point. Indeed, the qPCR experiments (Fig. 1b), monitoring of root colonization by plating of surface-sterilized plant sections (Supplementary Fig. 1b-c) and the plant infection assays (Supplementary Fig. 1a) were all performed in the same experimental setup, i.e. dip-inoculation of tomato plants grown in vermiculite. This is now clearly stated in the Figure legends and Methods section.

R2: Line 40: What do you mean by reference strain? Do you mean this is the type strain of the *forma specialis lycopersici*?

Response: The Fol isolate 4287 was the first strain of *F. oxysporum*, whose genome was sequenced (Ma et al. 2010, Nature) and since then has served as a reference isolate for studying this fungal pathogen.

R2: Lines 142 ff. Your only evidence for the fungal hyphae being used as a vector by bacterium is the microscopic in vitro assay shown in Fig. 3. Whether this results in a faster and easier

colonization of plant roots by the bacteria remains unknown. Similarly all chemotaxis, plant protection and acidification experiments were also performed in vitro (I regard vermiculite as a nearly-in vitro situation) The situation in nature might be completely different. Please consider this in your discussion. Did you ever investigate these interactions under more natural conditions? Did you ever perform the biocontrol experiment in soil? To my opinion an experiment is lacking investigating whether the bacterium reaches the plant earlier (e.g. if inoculated at a certain distance of the plant stem) and the root colonization is established earlier in presence of the fungus.

Response: We agree with the reviewer on the need to study this interaction in natural soil. Therefore, we repeated all key experiments in soil obtained from an agricultural setting, which was either untreated or sterilized. These include the plant mortality experiments (Fig. 1f; Supplementary Figs. 3g and 4a-b) as well as the pH/gluconic acid measurements (Fig. 2e). Moreover, we quantified Ra36 colonization of tomato roots upon inoculation at 5 cm distance from plant hypocotyls, into untreated or sterilized soil which either contains or lacks Fol (Fig. 4e). Importantly, under this setting the Ra36 wild type strain, but not the *fliC* deletion mutant impaired in flagellar motility, were able to colonize tomato roots more efficiently in presence of Fol, suggesting that Ra36 uses Fol hyphae as a vector in natural soil conditions.

R2: Material and method section: it is very hard to figure out which methods were used for which figure (subfigure). Please indicate this in the method section.

Response: In the revised version, we have restructured the Methods section to clarify this issue. We have also been careful to use the same terminology in the methods section and in the figure legends in order to facilitate understanding for the reader.

R2: LL 210-22 and LL270-282 combine and make a separate aPCR chapter. Description of qPCR conditions are simply missing.

Response: We thank the reviewer for this helpful suggestion. The information on Real-Time qPCR has now been merged with that of qualitative assessment of fungal and bacterial colonization in a new paragraph "Detection of Fol and Ra36 in tomato plant tissues". The conditions used for qPCR to quantify fungal and bacterial biomass in plant roots have now been fully described.

R2: Line 481. I don't understand the colors described in the legend. As it is indicated here, the purple area is the soil, but it surely is not. Isn't it the liquid film surrounding the hyphae? Please also indicate the hyphae and the root in the picture. And what are the orange circles? Are these soil particles? In general, I don't think this picture is very informative and would omit it.

Response: We agree with this appreciation and have removed the Figure.

R2: Figure 1: the letters of subfigs g and f should be exchanged to keep a an order from top to down and from left to right (1f should be the figure bottom left and 1g the figure bottom right).

Response: Most of the Figures have been substantially changed in the revised version, in order to accommodate the considerable amount of new experimental data. In doing so, we have also taken into account the suggestion of Reviewer 2.

R2: Figure 2 and extended data Fig 3a and c: . You indicate that * means sign. different at $P < 0.0001$, but based on what test? Please add statistics.

Response: The statistical tests used have now been included in all the Figure legends, as well as and in a separate Supplementary Data file (Supplementary Data 1).

R2: Fig. 3. I am not sure, whether all these figs should be shown in the main text. e.g Fig 3c could be moved to the extended data.

Response: We believe that the results shown in Fig. 3c (now Fig. 4c) are highly informative and therefore have opted for maintaining them in the main Figure.

R2: Extended data Fig 2: exchange letters h and g, it is confusing that g is the figure bottom right and h the one bottom left.

Response: Most of the Figures have been substantially changed in the revised version, in order to accommodate the considerable amount of new experimental data. In doing so, we have also taken into account the suggestion of Reviewer 2.

R2: 3g: Neither in the figure nor in the legend there is an indication which plates were treated with water and which with Fol conidia. Please add.

Response: We thank the reviewer for pointing out this mistake. The labels H₂O and Fol have now been added to the Figure.

R2: Comments on experimental setups:

Fig 1a: and 1g, extended Data Fig. 1i and 1k: How many seedlings per treatment were used? Show at least one repetition of the experiment in the extended data or the significances of the repetitions in extended data tables 1-4 (it would also be helpful for the reader to indicate in the extended tables to which Figs these statistics belong)

Response: In the revised version, the number of seedlings used for each treatment is indicated in the figure legends. As suggested by the reviewer, one additional repetition of each of the key experiments has been included (Supplementary Fig. 4). Moreover, all the significances and P values obtained for the statistical tests have been included in a separate excel file (Supplementary Data 1), where data are organized in separate spreadsheets according to Figures and panels. Additionally, source data for all relevant experiments are provided in a separate Source Data file.

R2: Fig 1b and 3e: As I can extract from the Figure legend, the data shown in this Figure are averages of three technical not biological replicates. In this case, it is absolutely necessary to show results of experimental repetitions, since in fact what you show here are data from only a single biological replicate per treatment.

Response: Indeed, the data shown in Figs. 1b and 3e (now Figs. 1b and 4e) are averages \pm SE of three biological replicates. For each biological replicate, the data represents the mean of 5 individual plants. We have revised the figure legends and the Methods section to clarify this.

R2: Fig 1f: Is the methods described in lines 257-262 the one for this figure? What is shown in this Figure? Results of one experiment? What are these averages based on? How many replicates? Show one repetition of experiment in extended data.

Response: The methods relating to Fig 1f (Fig. 2d in the revised version of the manuscript) are described in lines 305-310 and 319-322. Data shown are averages \pm SD of four biological replicates. The data from each biological replicate represents the mean of three technical replicates. We have revised the figure legends and the Methods section to clarify this.

R2: Fig. 2 b-2e, see comment Fig 1f. What are the averages shown here based on? Are these averages of three independent experiments with only one replicate per experiment? Or averages of several replicates of one of the three experiments? If yes, how many replicates were used to create these averages?

Response: Data shown in Fig. 2b-e (now Fig. 3b-e) are averages \pm SD of three to four biological replicates. The data from each biological replicate represents the mean of two technical replicates. We have revised the figure legends and the Methods section to clarify this.

R2: Figure 3 e: see comments on Fig 1b.

Response: See response above.

R2: Extended data Fig 3a, d, e and f, see comments on Fig 1f, are results of 1 of the 3, respectively 4 independent expts shown? If yes how many replicates were used?

Response: Data shown in the Extended data Fig 3a,d,e,f (now Supplementary Fig. 3a,c-e) are averages \pm SD of six biological replicates. The data from each biological replicate represents the mean of three technical replicates. We have revised the figure legends and the Methods section to clarify this.

Reviewer #3 (Remarks to the Author):

Major points

R3: It is still not clear how secretion of GclA by bacterial endophytes overcomes fungal-mediated alkalization, despite the fact that fungal pathogens should also secrete peptides that induce alkalization at the same timing. Is this just because the root endophyte bacteria secrete too much GclA (bacterial dose dependence)? Alternatively, does secretion of GclA (and/or endophytes) also inhibit fungal processes that are necessary for alkalization (Masachis, S. et al. Nature Microbiology)?

Response: Thank you for bringing up this interesting point. While at present we cannot exclude that GlcA secretion could inhibit expression of the Fol alkalizing peptide F-RALF (Masachis et al. 2016, Nat Microbiol), we believe that the new experimental data from co-inoculation of Fol with increasing concentrations of Ra36 strongly point towards a bacterial dose dependent effect (see new Fig. 1a)

R3: It is necessary to provide more detailed mechanisms by which bacteria are attracted by *F. ox* hyphae. For this purpose, it might be useful to test whether bacteria attraction to fungal hyphae (especially at the hyphal tip) is related with fungal-mediated alkalization. To test this possibility, using fungal mutants that cannot secrete alkalization peptides would be useful (Masachis, S. et al. Nature Microbiology).

Response: While this is an interesting question, the use of Fol f-ralf mutants in the experimental setting used here (standard in vitro capillary chemotaxis assay) does not provide a conclusive answer, because the alkalizing activity of the F-RALF peptide strictly depends on the presence of the plant (via interaction with the plant receptor kinase FERONIA, see Masachis et al. 2016, Nat Microbiol).

R3: Related with this, it might be also useful to test whether bacteria movement to host roots through hyphae depend on secretion of GlcA by using bacteria mutants without GlcA secretion (*gcd-*). As the concept that bacteria use fungal hyphae as a highway have been reported already (e.g., Yuanchen Zhang et al., Nature communications 2018), I think it would necessary to provide more mechanistic details in some ways.

Response: We thank the reviewer for this valuable suggestion. In the revised version, we have now included the Ra36 *gcd-* mutant in the chemotaxis assay and found that GlcA secretion is not required for chemoattraction of R36a towards Fol-inoculated tomato roots or towards alkaline pH (see Fig. 3b, e).

R3: It seems that only one *gcd-* mutant line is described in this work. It is necessary to test multiple mutants or the complementation lines to validate that defects in the gene is responsible for the phenotype shown in at least some of the figures.

This point was raised by reviewers #2 and #3. We agree with both reviewers on the need to test either complemented lines or multiple mutants of the *gcd* gene at least in the key experiments. In the revised version of the manuscript, we included the results for an additional independent knockout mutant, as well as for an ectopic transformant in which the integration of the gene knockout cassette occurred in a different genomic region. As expected, the second knockout mutant was also impaired in production of GlcA, extracellular acidification and control of Fol infection, whereas the ectopic transformant was similar to the Ra36 wild type strain (Fig. 2d and Supplementary Fig. 1a). An analogous approach was used to validate that the effect of deletion of the *fliC* gene on bacterial motility (Supplementary Fig. 2h).

R3: It would be also interesting to test whether bacteria flagella is necessary to efficiently move on fungal hyphae and to exhibit bacteria-mediated biocontrol activities. Although I predict that flagella is necessary to move efficiently on fungal hyphae, it might be not essential for the bacteria biocontrol activities when bacteria are directly inoculated onto tomato roots (like Fig.1).

Response: Thank you for this excellent suggestion. We agree that including a flagellar mutant of Ra36 is of high interest to understand the contribution of motility to biocontrol. We therefore generated Ra36 mutants lacking the orthologue of the *fliC* gene encoding a structural flagellin essential for flagellum function and bacterial motility (Blair 1995. *Annu. Rev. Microbiol.* 49:489–522). As expected, the *fliC*⁻ mutant failed to exhibit chemotaxis towards any of the tested chemoattractants, including alkaline pH (Fig. 3b, e). Moreover, colonization of tomato roots by the *fliC*⁻ mutant was not enhanced in the presence of Fol in the soil, which is in stark contrast to the Ra36 wild type strain (Fig. 4e). Importantly, when Ra36 was inoculated directly on the roots, FliC-mediated bacterial movement was not required to provide protection of tomato plants against Fol vascular wilt (Supplementary Figs. 3g and 4b).

Minor points

R3: The interactions shown here does not seem to provide any benefits to fungal-side. Do you think the type of interactions is really sustainable in natural ecosystems? Is this related with the fact that GlcA treatment improves *F. ox* growth in vitro (Extended Fig 2h)?

Response: We do not interpret the results shown in Supplementary Fig. 2g as a significant positive effect of GlcA on colony growth of *Fol*. That said, it cannot be discarded that GlcA-mediated extracellular acidification could somehow benefit the fungus in a yet unknown way. Currently we do not have sufficient experimental data to answer this interesting question.

REVIEWERS' COMMENTS:

Reviewer #1 (Remarks to the Author):

I commend the authors on addressing all of the major comments from myself and the other reviewers. The manuscript is definitely stronger because of the additional data and experiments included. The authors have addressed all of my major concerns with new data and rewriting.

A new issue I noticed during review of the revision is that one image appears to be duplicated in Figure 2c. In the Fol+ Ra36 treatment (3rd pair of images down), the two images are exactly the same even though 72 hours has passed. It is impossible for the plastic wrap used on the Petri dish to be in the exact same location following 72 hours of growth (note how the wrap position changes in all other photos). If this image duplication was done intentionally, then I am concerned about the credibility of other aspects of this manuscript. If it was done accidentally, the authors need to fix this.

Minor:

In Figure 2a and 2d, the authors need to indicate what the #3 and #5 mean (independent isogenic deletion mutants, I believe).

Reviewer #3 (Remarks to the Author):

I have carefully read through all the parts of the manuscript and found that the authors have answered majority of the concerns raised by the reviewers.

In this round, to further improve the manuscript, I would like to point out few minor points that worth considering.

1. One of the most important figures in this study should be figure 4e. But the bottom graph representing Fol biomass is not clear.

Changing choice of unit width in Y axis might help to solve this issue. Please consider a way to make this graph clear.

2. Related to figure 4e, it is interesting to see that Ra36 without FliC still inhibits Fol growth in roots. I suppose that this is because Ra36 without FliC inhibits hyphal growth at the contact site (5 cm distance from the plant hypocotyl) via GluA. But to make this point clear, it would be useful to test whether Ra36 without FliC inhibits hyphal aggregates of Fol as similar to WT strain (like Fig S1e).

3. Line 82, from the survival rate that only shows live or dead of plant individuals (Fig.1a), it is not possible to conclude that Ra36 did not cause any detectable symptoms in tomato plants. If the authors would like to mention this, it would be necessary to measure the presence or absence of disease symptoms or measure plant shoot fresh weight as indicator of plant growth in this experimental setting. Alternatively, it would be necessary to change the phrase.

Response to reviewers:

We are grateful to the two expert reviewers for their positive comments and constructive criticisms. In the revised version of the manuscript we have now addressed all the points raised by the reviewers. Below is a list of detailed point-to-point responses:

Reviewer #1 (Remarks to the Author):

R1: I commend the authors on addressing all of the major comments from myself and the other reviewers. The manuscript is definitely stronger because of the additional data and experiments included. The authors have addressed all of my major concerns with new data and rewriting. A new issue I noticed during review of the revision is that one image appears to be duplicated in Figure 2c. In the Fol+ Ra36 treatment (3rd pair of images down), the two images are exactly the same even though 72 hours has passed. It is impossible for the plastic wrap used on the Petri dish to be in the exact same location following 72 hours of growth (note how the wrap position changes in all other photos). If this image duplication was done intentionally, then I am concerned about the credibility of other aspects of this manuscript. If it was done accidentally, the authors need to fix this.

Response: We are grateful to the reviewer for the positive appreciation of our work and thank him/her for pointing out our mistake, which occurred during the figure panel assembly. We have now replaced the duplicated image of the Fol+ Ra36 treatment in the Figure panel 2c with the correct image obtained from the same petri dish after 72 h incubation.

R1: Minor: In Figure 2a and 2d, the authors need to indicate what the #3 and #5 mean (independent isogenic deletion mutants, I believe).

Response: We fully agree with the reviewer. We have now added this information in the Figure 2 legend, as well as in the legends of Supplementary Figs. 1 and 2.

Reviewer #3 (Remarks to the Author):

R3: I have carefully read through all the parts of the manuscript and found that the authors have answered majority of the concerns raised by the reviewers. In this round, to further improve the manuscript, I would like to point out few minor points that worth considering.

1. One of the most important figures in this study should be figure 4e. But the bottom graph representing Fol biomass is not clear. Changing choice of unit width in Y axis might help to solve this issue. Please consider a way to make this graph clear.

Response: We thank the reviewer for the helpful suggestion. In the revised version of the manuscript we have now changed unit width on the Y axis and vertically stretched the graph representing Fol biomass to increase bar sizes.

R3: 2. Related to figure 4e, it is interesting to see that Ra36 without FliC still inhibits Fol growth in roots. I suppose that this is because Ra36 without FliC inhibits hyphal growth at the contact site (5 cm distance from the plant hypocotyl) via GlcA. But to make this point clear, it would be useful to test whether Ra36 without Flic inhibits hyphal aggregates of Fol as similar to WT strain (like Fig S1e).

Response: As stated in our previous Responses to the reviewers and in the manuscript (lines 95-96 and line 140), the results shown in Supplementary Fig. 1e and Supplementary Fig. 2f fail to show a significant negative or positive effect of either Ra36 or GlcA on colony growth of Fol. Thus, we consider it is highly unlikely that the Ra36 FliC⁻ mutant should have any inhibitory effect on Fol growth, as suggested by the reviewer. We rather propose that the reduced concentration of Fol in tomato roots upon co-inoculation with Ra36 FliC⁻ (compared to Fol alone) is a result of the FliC⁻ mutant still being able to acidify the extracellular space through GlcA secretion. We speculate that GlcA might diffuse from the inoculation site to the proximity of the plant root and cause some residual acidification to inhibit Fol virulence functions, including aggregation. In line with this, we found that the Ra36 FliC⁻ mutant strongly inhibits Fol aggregation similar to the wt strain when co-inoculated in the same well (see new image added in Supplementary Fig. 1g) and reduces fungal virulence when dip-inoculated directly on the plant root (Supplementary Fig. 3g). By contrast, the inhibitory effect of the FliC⁻ mutant on Fol colonization is much weaker compared to the wt strain when inoculation is performed at a 5 cm distance from the plant hypocotyl.

To clarify the concept raised by the reviewer, we have now included an additional image showing Fol hyphal aggregation in the presence of the Ra36 FliC⁻ strain (Supplementary Fig. 1g).

R3: 3. Line 82, from the survival rate that only shows live or dead of plant individuals (Fig.1a), it is not possible to conclude that Ra36 did not cause any detectable symptoms in tomato plants. If the authors would like to mention this, it would be necessary to measure the presence or absence of disease symptoms or measure plant shoot fresh weight as indicator of plant growth in this experimental setting. Alternatively, it would be necessary to change the phrase.

Response: We thank the reviewer for his/her helpful comment. We have rephrased the indicated sentence (now Line 86) as follows: "When inoculated alone, Ra36 did not cause any mortality in tomato plants (Fig. 1a)."